# FLOP: Tasks for Fitness Landscapes Of Protein wildtypes

**Peter Mørch Groth**[1,2], **Richard Michael**[1],
**Jesper Salomon**[2], **Pengfei Tian**[2], **Wouter Boomsma**[1]
[1]Department of Computer Science, University of Copenhagen
[2]Bioinformatics & Design, Enzyme Research, Novozymes
{petergroth,richard.michael,wb}@di.ku.dk
{pmg,jrsx,pfi}@novozymes.com

## Abstract

Protein engineering has the potential to create optimized protein variants with improved properties and function. An initial step in the protein optimization process typically consists of a search among natural (wildtype) sequences to find the naturally occurring proteins with the most desirable properties. Promising candidates from this initial discovery phase then form the basis of the second step: a more local optimization procedure, exploring the space of variants separated from this candidate by a number of mutations. While considerable progress has been made on evaluating machine learning methods on single protein datasets, benchmarks of data-driven approaches for global fitness landscape exploration are still lacking. In this paper, we have carefully curated a representative benchmark dataset, which reflects industrially relevant scenarios for the initial wildtype discovery phase of protein engineering. We focus on exploration within a protein family, and investigate the downstream predictive power of various protein representation paradigms, i.e., protein language model-based representations, structure-based representations, and evolution-based representations. Our benchmark highlights the importance of coherent split strategies, and how we can be misled into overly optimistic estimates of the state of the field. The codebase and data can be accessed via https://github.com/petergroth/FLOP.

## 1 Introduction

The goal of protein engineering is to optimize proteins towards a particular trait of interest. This has applications both for industrial purposes and drug design. There is clear potential for machine learning to aid in this process. By predicting which protein sequences are most promising for experimental characterization, we can accelerate the exploration of the "fitness landscape" of the protein in question [1]. Regression of functional landscapes is challenging for multiple reasons. Typically a data-scarce problem, careful considerations of the experimental setup are required to avoid inadvertent data leakage. Concerning the functional landscapes of naturally occurring (also known as *wildtype*) proteins, the pairwise amino acid sequence identities can often vary significantly with some proteins differing by only a single amino acid while others might be less than ten percent similar. High-throughput experimental techniques are improving the data scarcity issue, while underlying structure typically exists in the datasets allowing for supervised learning despite the intrinsic challenges.

Submitted to the 37th Conference on Neural Information Processing Systems (NeurIPS 2023) Track on Datasets and Benchmarks. Do not distribute.

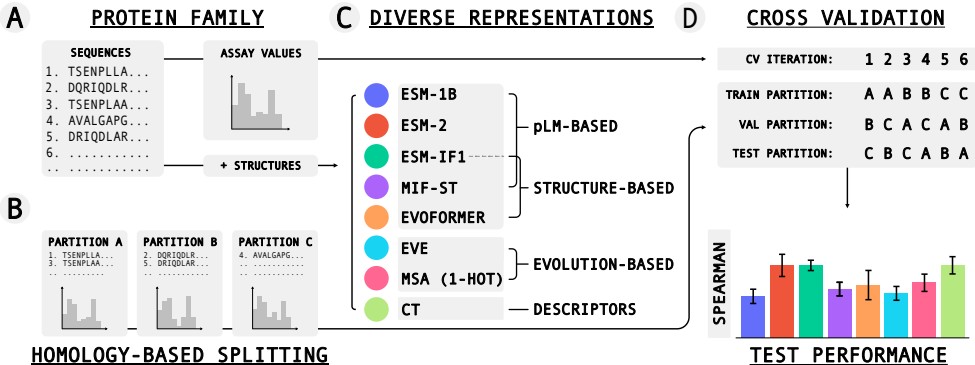

Figure 1: Schematic over dataset splitting, representations, and cross-validation process. A: A dataset with sequences from a single protein family and corresponding assay values is curated. B: A stratified sequence identity splitting procedure generates partitions A, B, and C, which are (1) homologically different from each other, (2) contain similar number of sequences, and (3) match the full dataset's target distribution. C: Eight types of protein representations are computed. D: Cross-validation using a random forest regressor is applied to obtain mean values and standard errors on the test partitions.

The optimization process of proteins and enzymes can typically be divided into multiple stages. An often employed initial step is to search for promising candidates among wildtype proteins, resulting in a set of proteins with desirable properties. We will refer to this as the *wildtype discovery* phase [2]. Since we are typically optimizing for a specific trait, we may often limit this initial exploration to a particular protein family, where the members share an evolutionary history which has resulted in a similar function. The selected set of wildtype proteins will then form the basis for a second phase in the engineering process: localized optimization, where novel variants of the wildtype proteins are examined through various assays [3]. Sometimes, the wildtype discovery phase is not only carried out once as several rounds might be required before an initial suitable candidate is found. Additionally, the resulting candidate might prove insufficient at a later stage of protein engineering, where conditions such as temperature are altered or where stress-factors are introduced.

In recent years, we have seen considerable efforts in defining benchmarks to help the machine learning community make progress in this field. However, these efforts have primarily focused on the second stage, i.e., variant effect prediction, where a dataset consist of thousands of variants from a single wildtype. In this paper, we argue for the importance of establishing well-defined benchmark tasks for the first stage as well. We present three challenging tasks and a careful analysis of the experimental design, demonstrating how poor choices can lead to dramatic overestimation of performance.

We conduct our experiments using a variety of fixed-size protein representations: sequence-based embeddings obtained through protein language models, structure-based representations from folding and inverse folding models, evolution-based representations obtained from multiple sequence alignments, as well as simple biologically-motivated sequence descriptors. In addition to the supervised approach, we include four zero-shot predictors to showcase a simpler approach to the task of identifying promising candidates. We show that the choice of representation can greatly affect the downstream predictive performance, and we therefore argue that more progress can be made by constructing meaningful representations and not solely in the construction of complex prediction models. Given the oftentimes limited dataset sizes, we therefore rely on a random forest regressor.

## 2 Related work

Benchmarks play an important role in driving progress in protein-related prediction tasks. The most well-known is perhaps the rolling CASP benchmark, which is arguably responsible for the recent breakthroughs in protein structure prediction [4–6]. For the prediction of protein stability and function, several studies have curated relevant experimental datasets for use as benchmarks. The TAPE benchmark was an early such example designed to test protein sequence representations on a

set of diverse downstream tasks [7]. Two of these tasks were related to protein engineering: stability prediction on variants of a set of 12 designed proteins [8] and characterization of the functional landscape of green fluorescent protein [9]. The PEER benchmark [10] expanded on the TAPE benchmark with many additional tasks. This included prediction of $\beta$-lactamase activity [11], and a binary solubility classification task on a diverse set of proteins. Focusing entirely on variant effects, the recent ProteinGym benchmark has assembled a large set of Deep Mutational Scanning (DMS) assays and made them available as substitution and insertion-deletion prediction tasks [12]. While the above all consider protein sequence inputs, the recent Atom3D benchmark [13] presents various prediction tasks using 3D structure as input, including predicting amino acid identity from structural environments (for general proteins), and mutation effects on protein binding, using data originating from the SKEMPI database [14, 15].

Most closely related to this current paper is the FLIP benchmark, which dedicates itself to the prediction of functional fitness landscapes of proteins for protein engineering [16]. FLIP introduces three tasks: one on the prediction of protein stability of wildtype proteins (distributed over many families) using data from the Meltome Atlas [17], and two tasks focused on mutations at specific sites of proteins GB1 [18] and AAV [19]. While the FLIP benchmark is of great value for protein engineering, there are key characteristics which make it unsuitable for wildtype discovery, e.g., the use of the Meltome Atlas, which consists of thousands of sequences from different organisms spanning many different protein families. The sequences in the GB1 dataset only have mutations at four fixed positions while the sequences in the AAV dataset only contain 39 mutation sites, both of which corresponds to mutations at less than 10% of the full-length proteins. Such datasets with very local fitness landscapes are not generalizable enough for wildtype discovery.

Most functional tasks in current benchmarks are thus concerned with protein sequences that are derived from a single wildtype sequence by one or more mutations. Characterizing the functional effects of such variants is critical for protein engineering. However, before engaging in the optimization process itself, it is important to select meaningful starting points. As a natural complement to the FLIP benchmark, we therefore present a novel benchmark titled FLOP. The tasks we present are the characterization of functional landscapes of wildtype proteins.

Our curated datasets all consist of functionally characterized wildtype sequences. For each dataset, we limit ourselves to a single family, and define our tasks as regression problems on the functional assay values. While mutational fitness landscape datasets are relatively abundant, few published datasets exist where the global fitness landscapes of wildtype proteins from single families are examined. This imposes limitations in the number and sizes of available datasets which are suitable for our considered problem. Given the low-data regime, the focus of our benchmark is thus to find representations of the protein input that makes few-shot or even zero-shot learning feasible. As a point of departure, we provide a set of state-of-the-art embeddings, reflecting different protein modalities.

## 3 Experimental setup

The domain we explore in this work is characterized by data scarcity, requiring special care in the design of the experimental setup. Figure 1 shows an overall schematic of the benchmarking process.

### 3.1 Dataset splitting

With the proliferation of large datasets and computationally demanding models, a common learning paradigm in machine learning is to rely on hold-out validation, whereby fixed training, validation, and testing sets are randomly generated. This method has several serious limitations when applied to biological datasets of limited sizes. Firstly, randomly splitting a dataset assumes that the data points are independent and identically distributed (i.i.d.). This is however not the case for members of a protein family which share common ancestors, leading to potential data leakage if protein sequences that are close in evolutionary space are placed in separate splits. Secondly, when splitting small datasets for a hold-out validation approach for supervised learning, the target values might not be well-balanced, resulting in dissimilar target distributions thus leading to bias and poor generalizability.

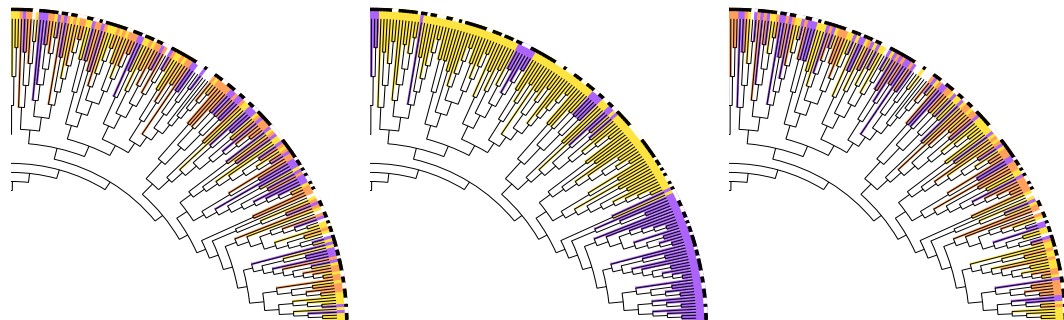

Figure 2: The same segment of a phylogenetic tree for the PPAT dataset. Branch color corresponds to its CV partition, while the outermost ring shows the target values (black indicates high and white indicates low values). The segments highlight the diversity found in wildtype protein families. Left: entries are colored according to the prescribed dataset splitting procedure which allows learning across subfamilies (indicated by the mix of colors). Middle: entries are colored by a clustering approach leading to wide regions, inhibiting learning across subfamilies. Right: entries are randomly assigned a color. While similar to the leftmost scheme, the random coloring allows near identical sequences to be placed in separate partitions leading to excessive data-leakage.

To handle these potential issues, we rely on a sequence identity-based, stratified cross-validation procedure ensuring that (1) partitions are generated such that any two proteins occurring in different partitions are guaranteed to be different at a pre-set homology cut-off, (2) cross-validation (CV) minimizes the potential bias which might occur during hold-out validation, (3) the target distribution is reflected by the generated partitions via stratification on discretized target values, and (4) the number of sequences in each partition is similar to reduce the variance.

To generate these high-quality data partitions, we use the four-phase procedure described in [20] and implemented in the GraphPart framework [21] to create three label-balanced partitions for each dataset. We begin the procedure from an initial sequence identity threshold, and increase the threshold until the generated partitions are of sufficient sizes (i.e., at least 25 % of sequences in all three partitions). The stratification is achieved by creating a binary label which indicates whether a protein has low or high target value, e.g., by fitting a two-component Gaussian mixture model. For the dataset-specific stratification boundaries, see Section A in the supplementary materials.

Figure 2 shows the same segment of a phylogenetic tree of the curated PPAT dataset, showing the evolutionary relationship between sequences. Large versions of the trees can be found in Section E. The colors indicate which CV partition each sequence belongs to while the black and white squares in the outer ring indicate the stratification labels. The segments show the diversity encountered in wildtype protein families. The left segment is colored by our splitting procedure and shows that it manages to create diverse partitions spanning the entire evolutionary tree to allow learning across protein subfamilies. The middle segment is colored by an MMseqs [22] clustering approach, leading to contiguous areas inhibiting learning across subfamilies. The entries in the rightmost segment are randomly assigned a color, corresponding to random splitting. While similar to the leftmost scheme, the random coloring allows near-identical sequences to be placed in separate partitions leading to excessive data-leakage.

## 3.2 Representations

To accurately reflect the current paradigms of state-of-the-art protein representations, we choose representatives from three main categories, the dimensionalities of which can be found in Section G in the supplementary materials.

Protein language models (pLMs) that are trained on hundreds of millions of protein sequences in an unsupervised fashion have been proven to be competitive for a multitude of tasks including supervised prediction of protein properties, residue contact prediction, variant effect prediction [16, 23–26], etc. We here choose the popular ESM-1B [24] and the more recent ESM-2 models [27]. To fix the

dimensionality for proteins of different lengths, we perform mean-pooling over the residue dimension. This operation is likely to filter out information encoded along the protein sequence and more optimal approaches will likely yield more informative representations and thus higher predictive performance (see Table 2 in [28]). Constructing fixed-size embeddings from sequences of variable lengths is however nontrivial and considered out of the scope of this study.

The second category we include is structure-based. We extract embeddings from the Evoformer-modules while folding proteins with AlphaFold2 [29] via ColabFold [30], which have been shown to perform well for structure-related prediction tasks [31]. Using the predicted structures, we then extract embeddings from the inverse-folding model ESM-IF1 (also known as the GVP-GNN) [32] which incorporates a pLM and graph neural network architecture. We similarly use embeddings from the MIF-ST model, which is an inverse folding model leveraging a pretrained convolutional pLM [33]. As with the pLMs, we apply mean pooling to achieve sequence-level embeddings.

The third category is evolution-based. As a baseline, we will use a one-hot encoded multiple sequence alignment (MSA) over the proteins of interest [34–36]. Since the MSA is independent of labels, we enrich the unaligned sequence pools with additional members from the respective protein families using UniProt [37] and InterPro [38]. Given MSAs, models can be designed which leverage the evolutionary history of the protein family (e.g., EVE and related models [12, 39–42]). For each curated dataset, we train EVE [39] on the corresponding protein family and extract the latent representations. Technical details on the training procedure can be found in Section H.

In addition to these groups of advanced representations, we include compositional and transitional (CT) physicochemical descriptors for each protein sequence as a simple baseline, which relate to overall polarizability, charge, hydrophobicity, polarity, secondary structure, solvent accessibility, and van der Waals volume of each sequence as predicted using the PyBioMed library [43].

With the exception of the physicochemical descriptors, all included representations rely on models which have been pretrained on thousands to hundreds of millions of proteins. While it is possible that a number of the sequences in the curated datasets also belong to the training sets of these models (which by design is the case for the evolution-based approaches), we do not consider this to be a fatal form of data leakage as it purely pertains to the un- or self-supervised pretraining phases and is independent of the sequence labels.

### 3.3  Regression

The purpose of this benchmark is to provide a structured procedure to evaluate the predictive performance on downstream regression tasks given protein representations. We believe that larger prediction improvements can be achieved by focusing on developing novel protein representations rather than more complex regression models. Due to the low-N setting in which we operate, the training of large, complex models is practically inhibited, which is why we have chosen to rely on a random forest regressor. For each combination of the generated CV partitions, we perform a hyperparameter optimization on the current validation partition and evaluate the best-performing predictor on the current test partition. The experiments were also carried out using alternate regressors. See Sections N.1 and K for these results and all hyperparameter grids, respectively.

### 3.4  Zero-shot predictors

To investigate the efficacy of unsupervised learning on the curated datasets, we evaluate four zero-shot predictors. Using EVE, we evaluate the evidence lower bound (ELBO) by sampling and obtain a proxy for sequence fitness, analogous to the evolutionary index in [39]. Second and third proxies are obtained by evaluating the log-likelihood of a sequence conditioned on its structure using the inverse folding models ESM-IF1 [32] and ProteinMPNN [44]. The fourth zero-shot estimator is obtained by using Tranception [45] to evaluate the log-likelihood of each sequence. Details for the use of ProteinMPNN and Tranception can be found in Sections I and J in the supplementary materials.

Table 1: Summary of datasets and splits.

| | $N_{tot}$ | $N_A$ | $N_B$ | $N_C$ | Split %ID | Target | Median %ID | Avg. length |
|---|---|---|---|---|---|---|---|---|
| **GH114** | 55 | 20 | 18 | 17 | 0.55 | Activity | 0.46 | 268.8 |
| **CM** | 855 | 341 | 259 | 255 | 0.40 | Activity | 0.40 | 91.1 |
| **PPAT** | 615 | 182 | 234 | 199 | 0.55 | Fitness | 0.51 | 161.6 |

## 4 Datasets

The three curated datasets and the corresponding fitness landscapes are here motivated and described. Despite the scarcity of available datasets described in Section 2, the curated datasets are representative examples of wildtype discovery campaigns in terms of size and diversity. For additional curation details on each dataset including specific thresholds for stratified splitting, see Section A.

### 4.1 GH114

**Motivation.** Accurately identifying enzymes with the highest activities towards a specific substrate is of central importance during enzyme engineering. To achieve this, it is essential to ensure that assay observations are directly comparable [46]. This includes maintaining identical experimental assay conditions, including evaluating enzymes at the same concentrations and purity levels. However, purifying enzymes requires significant work and resources, often resulting in assays composed of fewer sequences, which are in turn of higher experimental quality.

**Landscape.** This dataset includes purified and concentration normalized natural glycoside hydrolase 114 (GH114) alpha-1,4-polygalactosaminidase enzymes and corresponding catalytic activity values [47] which will act as the target of interest. GH114 enzymes degrade the exopolysaccharide PEL, which provides structure and protection in some biofilms [48]. Having measurements of purified enzymes avoids issues with background effects from other enzymes in the recombinant host background. We provide a curated version of the GH114 dataset which, to our knowledge, has not been used in previous work for function prediction purposes.

### 4.2 CM

**Motivation.** Identification of enzymes with high catalytic activities is essential for enzyme engineering campaigns. However, predicting the activity level of enzymes using physics-based methods remains a great challenge [49]. Recent progress in high throughput screening allows the measurement of enzyme activity of sequences with high diversity, but with low experimental cost.

**Landscape.** This dataset contains the catalytic activity of chorismate mutase (CM) homologous proteins, as well as artificial sequences which follow the same pattern of variations (e.g., conservation and co-evolution) [50]. The artificial sequences generated by Monte Carlo simulations at low and medium temperatures match the empirical first-, second-, and higher-order statistics of the natural homologs, while also exhibiting comparable catalytic levels when experimentally synthesized. These sequence have therefore been included given the similarity in both sequence and fitness landscape. See Section A.3 for further details. We perform an additional filtering of the dataset prior to the splitting procedure by removing sequences with target values less than 0.42, corresponding to inactive proteins [50]. This task thereby assumes that a preceding classification procedure has been carried out. For completeness, we include benchmark results for the CM dataset when only the natural homologs were used (see Section N.4) and classification results before the filtering step (see Section M), which supports this last assumption.

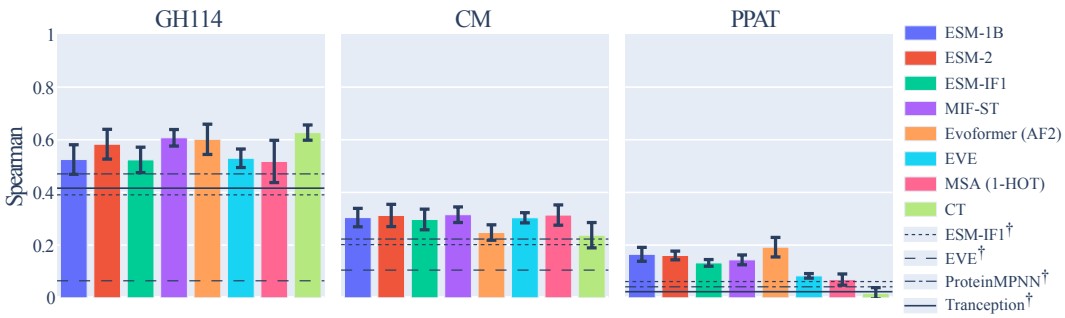

Figure 3: Average Spearman's rank correlation (and standard error) between predictions and targets over test partitions. Higher is better. [†]: Zero-shot correlations.

### 4.3 PPAT

**Motivation.** PPAT (phosphopantetheine adenylyltransferase) is an essential enzyme that catalyzes the second-to-last step in the CoA biosynthetic pathway. The target value for this prediction task is the fitness score, which reflects the ability of PPAT homologs to complement a knockout E. coli strain. The fitness of homologs can be affected by factors such as protein misfolding, mismatched metabolic flux, or environmental mismatches etc. [51].

**Landscape.** This dataset contains fitness scores of 615 different PPAT homologs obtained by a novel DNA synthesis/assembly technology, DropSynth, followed by a multiplexed functional assay to measure how well each PPAT homolog can rescue a knockout phenotype [51].

### 4.4 Summary

A summary of the curated datasets including their total sizes, partition sizes, between-partition sequence identity threshold (Split %ID), regression target, median pairwise sequence identity, and average sequence length can be seen in Table 1. Additional curation details are found in the supplementary materials (see Section A), while histograms of the regression targets for the three datasets as well as partition histograms can be seen in Sections C and D, respectively.

The low median sequence identity observed in Table 1 highlights the diversity – and difficulty – of wildtype datasets. Deep mutational scanning (DMS) datasets commonly used for variant effect prediction on average have median sequence identities greater than 0.99. For comparison, Section F shows both median, mean, and standard deviation for the curated datasets (see Table A1) contrasted to 48 tasks from the ProteinGym benchmark (see Table A2). A visual example highlighting the sequence diversity for a protein family compared to a DMS dataset can be seen in Figure 4 in [40]. The DMS data is localized to a small section of the protein family as it is composed of all single mutations of a single wildtype. Similarly, a DMS of a protein belonging to the PPAT family would all be positioned on a single branch of the phylogenetic tree in Figure 2.

## 5 Results

Spearman's rank correlation between the predictions and targets over the three datasets can be seen in Figure 3 and Table 2. The highest performing proteins are of interest making it the ranking and not the absolute predictions that indicate the performance, despite the regressors being optimized using the mean squared error. The RMSE can found in Table A4 in the supplementary materials.

**GH114** The performance on the GH114 dataset highlights some of the peculiarities encountered with small wildtype protein datasets. The collection of physicochemical descriptors (CT), which is simpler than its competitors, achieves the highest score. Slightly below it are the structure-informed MIF-ST and Evoformer representations, indicating a structural signal which is however not picked

Table 2: Benchmark results. Mean Spearman correlation and standard error using cross-validation. [†]: Zero-shot correlation on full datasets. Highest value and values within 1 SE are bold.

| | GH114 | CM | PPAT |
|---|---|---|---|
| ESM-1B | $0.52 \pm 0.06$ | $\mathbf{0.30} \pm 0.03$ | $\mathbf{0.16} \pm 0.03$ |
| ESM-2 | $\mathbf{0.58} \pm 0.06$ | $\mathbf{0.31} \pm 0.04$ | $\mathbf{0.16} \pm 0.02$ |
| ESM-IF1 | $0.52 \pm 0.05$ | $\mathbf{0.30} \pm 0.04$ | $0.13 \pm 0.01$ |
| MIF-ST | $\mathbf{0.61} \pm 0.03$ | $\mathbf{0.32} \pm 0.03$ | $0.14 \pm 0.02$ |
| Evoformer (AF2) | $\mathbf{0.60} \pm 0.06$ | $0.25 \pm 0.03$ | $\mathbf{0.19} \pm 0.04$ |
| EVE | $0.53 \pm 0.04$ | $\mathbf{0.30} \pm 0.02$ | $0.08 \pm 0.01$ |
| MSA (1-HOT) | $0.52 \pm 0.08$ | $\mathbf{0.31} \pm 0.04$ | $0.07 \pm 0.02$ |
| CT | $\mathbf{0.63} \pm 0.03$ | $0.24 \pm 0.05$ | $0.02 \pm 0.02$ |
| ESM-IF1[†] | 0.39 | 0.20 | 0.06 |
| EVE[†] | 0.06 | 0.11 | $-0.01$ |
| ProteinMPNN[†] | 0.47 | 0.22 | 0.04 |
| Tranception[†] | 0.42 | $-0.05$ | 0.02 |

up by the ESM-IF1 embeddings. While the CT representation achieves the highest mean value, several others are within one standard error, giving no clear advantage to neither complex nor simple models. While all supervised approaches beat the zero-shot predictors, ProteinMPNN, Tranception and ESM-IF1 likelihoods correlate well with the targets.

**CM** The second prediction task can be considered more challenging given the results, despite the comparatively large size of the CM dataset. While similar to the first task, an abundance of data is not sufficient to increase the downstream capabilities if it comes at the cost of potentially noisier measurements, as compared to the concentration normalized GH114 dataset. Most representations fall within one standard error of the top performer such that, once again, no representation paradigm has a clear advantage.

**PPAT** The most challenging task of the datasets, the results on the PPAT task show different behaviour. The evolutionary signal, i.e., the amount of information which can be learned from evolutionary homologs, is weak as indicated by the low correlations from the one-hot encoded MSA and from EVE (both in the supervised and zero-shot settings as per Table 2). Furthermore, the physicochemical descriptors fail to correlate – as do the remaining zero-shot predictors. The pLM and structure-based representations achieve the highest scores with the Evoformer embeddings coming out slightly ahead.

# 6 Ablation study

The dataset splitting procedure and benchmark tasks have been carefully constructed to ensure reliable estimates of model performance. In this section we show three ablation studies – one for each dataset – whereby different choices of task-structuring might lead to great over-estimations of performance. The results can be seen in Table 3 and in Figure A8. The $\Delta$ columns in the table indicate differences to the benchmark results, where a positive/green value indicates *better* performance during ablation, i.e., over-estimation.

**Hold-out validation** For GH114, we perform hold-out validation by arbitrarily designating the three generated partitions as training, validation, and test sets and running the experiment only once. The correlations are significantly different to the benchmark results, with the ESM-2 correlation decreasing by 50 %. With no systematic pattern and decreased nuance given the lack of errorbars, it is easy to draw incorrect conclusions.

As the data-leakage between partitions has been controlled via the splitting procedure, the partitions are different from each other up to the sequence identity threshold. This implies that a model might

Table 3: Ablation results. Spearman correlation. *: Hold-out validation, **: Regression on both active and inactive proteins, ***: Repeated random splitting. Δ shows difference to benchmark results. Highest value and values within 1 SE are bold.

| | GH114* | Δ | CM** | Δ | PPAT*** | Δ |
|---|---|---|---|---|---|---|
| ESM-1B | 0.36 | −0.16 | 0.64 ± 0.01 | +0.33 | **0.23** ± 0.05 | +0.07 |
| ESM-2 | 0.39 | −0.20 | **0.66** ± 0.01 | +0.35 | 0.18 ± 0.02 | +0.02 |
| ESM-IF1 | 0.46 | −0.06 | 0.58 ± 0.01 | +0.28 | 0.19 ± 0.01 | +0.06 |
| MIF-ST | 0.62 | +0.01 | 0.60 ± 0.02 | +0.28 | **0.24** ± 0.04 | +0.09 |
| Evoformer (AF2) | 0.64 | +0.04 | 0.57 ± 0.01 | +0.33 | **0.24** ± 0.02 | +0.04 |
| EVE | 0.38 ± 0.04 | −0.15 | 0.62 ± 0.00 | +0.31 | 0.16 ± 0.01 | +0.08 |
| MSA (1-HOT) | 0.50 | −0.02 | 0.61 ± 0.01 | +0.30 | 0.06 ± 0.06 | −0.01 |
| CT | **0.65** | +0.03 | 0.52 ± 0.01 | +0.28 | 0.13 ± 0.01 | +0.11 |

perform well on, e.g., only a subset of the partitions. Choosing which partitions to use for training, validation, and testing is (in this case) arbitrary and can thereby lead to misleading results. To avoid this pitfall, cross-validation is needed such that the average predictive performance on all combinations of partitions can be estimated. An analogue ablation study for the CM and PPAT datasets can be found in Section L.1, where similar conclusions can be drawn.

**Disregarding distinct target modalities**   For the CM dataset, we only included the active sequences in the benchmark. To demonstrate why, we have included the results of performing regression on both active *and* inactive sequences in the center of Table 3. These results are greatly overinflated compared to the benchmark results, with some representations more than doubling the correlation scores.

Regression performed on a dataset with a distinctly bimodal target distribution (such as the full CM dataset, see Figure C in the supplementary materials) can inflate the results significantly. The regressor is able to distinguish between the two target modalities, i.e., between the inactive cluster around 0 and the active cluster around 1, driving the ranking correlation to overly-optimistic values. The caveat to this preprocessing step is that it requires knowing the whether the proteins are active or not a priori, which assumes that a preceding classification-screening has been performed. The classification results of such a process can be seen in Section M.

**Random partitioning for cross-validation**   To illustrate why random splitting of wildtype protein datasets is ill-advised, we applied repeated random splitting to the PPAT dataset. This was done by randomly assigning sequences to training, validation, and testing partitions without any consideration of sequence similarity. Given the randomized partitions, the predictive performance using the selected representations was evaluated using cross validation. This was repeated a total of three times with different seeds. While the results look similar to the benchmark results, we do see an increase in performance across the board.

With random sampling, we risk placing very similar sequences in separate partitions, thereby allowing extensive data-leakage, where we are essentially testing on training/validation data, thus overestimating the predictive performance [52]. The results for this ablation study carried out on the GH114 and CM datasets can be found in Section L.2, where we can once again draw similar conclusions.

## 7   Discussion

The choice of representation greatly affects the downstream predictive capabilities, with no consistent, clear edge given by any of the three representation paradigms. For CM, a one-hot encoded MSA acts as an impressive baseline proving difficult to convincingly beat. For GH114, physicochemical descriptors are sufficient to achieve top performance, while the PPAT dataset benefits from the complex, structure-informed Evoformer embeddings. While the specific top-scoring representation fluctuates, the ESM-2 embeddings are consistently within one standard error and can thus be considered a relatively

consistent baseline for future experiments, where others occasionally underperform. For the three tasks, we see supervised learning outperforming zero-shot predictions, while the inverse-folding estimators however offer decent zero-shot approaches for two of three tasks.

Despite similar overall patterns, some results stand out, e.g., the comparatively high performance on the GH114 dataset and the low performance on the PPAT dataset. Variations in experimental conditions and techniques can introduce different levels of noise. The CM and PPAT datasets are derived from tests on supernatants with complex backgrounds with potential side-activities from impurities, whereas the GH114 dataset uses purified samples with less expected noise. This can be a potential reason for the comparatively high performance of the latter. As for the low performance on the PPAT dataset, the reason might lie in the target values: the GH114 and CM datasets both measure enzymatic activities while the PPAT dataset measures fitness. The overall performance disparities suggest that enzyme activities, rather than a more complex and assay-specific fitness value, are easier to model given the available protein representation paradigms. The stark contrast in performance between the concentration normalized GH114 dataset and both the CM and PPAT datasets indicates that higher quality datasets are of central importance to learn accurate fitness landscapes – more so than the number of labelled sequences.

# 8   Conclusion

In this work we have presented a novel benchmark which investigates an unexplored domain of machine learning-driven protein engineering: the navigation of global fitness landscapes for single protein families. Wildtype exploration can be viewed as a predominantly explorative phase of protein optimization, which precedes the exploitation phase comprised of the subsequent protein engineering. Often, limited resources are allocated to wildtype exploration since it is inherently costly and considered wasteful as it tends to produce many poor candidates. This is unlikely to change unless we find ways to improve our wildtype search strategy, which will require better predictions. We therefore consider the limited dataset sizes as an inherent condition and limitation in this domain. This makes the collection and curation of relevant labelled datasets challenging and also necessitates the design of careful learning schemes and model evaluation to ensure reliable estimates of generalizability while avoiding inadvertently overestimating the results. We anticipate that the creation of this new set of comprehensive family-wide datasets will facilitate and improve future model development and applicability in this domain.

Given the limited dataset sizes, our focus has been on transfer learning and zero-shot prediction. Our results show that the supervised approaches outperform the zero-shot approaches, but that no one representation or representation paradigm consistently outperforms the others. This could suggest that the employed representations are not sufficiently informative. A key limitation for a number of the included representations is that we obtained protein-level representations as averages over the protein length to arrive at fixed-length embeddings, which is known to be suboptimal [28]. We encourage the community to experiment with novel aggregation strategies and new representation designs to improve performance on our benchmark. It is also conceivable that general-purpose protein representation models might not by themselves be sufficient to convincingly improve on the proposed tasks. One can imagine that further improvements can be obtained using pretrained models fine-tuned on a protein family of interest – or by developing weakly-supervised representation models incorporating relevant properties that correlate with the function of interest (e.g., thermostability).

Although the performance of current baselines on some of our test-cases is fairly low in absolute terms, even low correlations can provide useful guidance on selecting wildtype protein starting points and can have measurable real-world impacts. Any further improvements will enhance the importance of wildtype exploration relative to the subsequent local optimization step. In silico screenings of potential wildtype candidates can be scaled efficiently compared to expensive, time-consuming in vitro assays, significantly reducing the early costs of future protein engineering campaigns. We hope that FLOP will pave the way for these developments.

## Acknowledgments and Disclosure of Funding

This work was funded in part by Innovation Fund Denmark (1044-00158A), the Novo Nordisk Foundation through the MLSS Center (Basic Machine Learning Research in Life Science, NNF20OC0062606), the Pioneer Centre for AI (DRNF grant number P1), and the Danish Data Science Academy, which is funded by the Novo Nordisk Foundation (NNF21SA0069429) and VILLUM FONDEN (40516).

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
