# Supplementary materials for FLOP:
# Tasks for Fitness Landscapes Of Protein wildtypes

**Peter Mørch Groth**[1,2], **Richard Michael**[1],
**Jesper Salomon**[2], **Pengfei Tian**[2], **Wouter Boomsma**[1]
[1]Department of Computer Science, University of Copenhagen
[2]Bioinformatics & Design, Enzyme Research, Novozymes
{petergroth,richard.michael,wb}@di.ku.dk
{pmg,jrsx,pfi}@novozymes.com

## A    Dataset details

The curated datasets are kept in `csv`-files with the following columns:

- `index`: Index for each protein.

- `name`: Unique name for each protein. This identifier maps directly to the file name for all representations. For example, the ESM-2 embedding for sequence `<seq_id>` from `<dataset>` can be found in `representations/<dataset>/esm_2/<seq_id>.pt`.

- `sequence`: Amino acid sequence.

- `target_reg`: The assay value/regression target.

- `target_class`: Binarized assay value for stratification.[1]

- `part_0`: 1 if sequence belongs to the first partition, 0 otherwise.

- `part_1`: 1 if sequence belongs to the second partition, 0 otherwise.

- `part_2`: 1 if sequence belongs to the third partition, 0 otherwise.

The curated file for `<dataset>` is placed in `data/processed/<dataset>/<dataset>.csv`. For details on data access, see Section A.1.

All structures were predicted with AlphaFold2 [1] using ColabFold [2] using five recycling runs with model version `alphafold2_multimer_v3` with early stopping at pLDDT of 90.0. The predicted structures can be found in the `data/raw/<dataset>/pdb` directory for each dataset.

We ask that references to the tasks in this paper include references to the original dataset authors.

### A.1    Dataset/code access

All code is accessible via the repository at `https://github.com/petergroth/FLOP`. The three curated dataset files can be found in the repository as three separate `csv`-files. All remaining files (including PDB-files, pre-computed representations, raw data files, etc.) can be found at `https://sid.erda.dk/sharelink/HLXs3e9yCu`. Additional details can be found in the repository.

---

[1]Tasks can alternatively be cast as classification problems by predicting this column instead, as was also done for the CM dataset.

Submitted to the 37th Conference on Neural Information Processing Systems (NeurIPS 2023) Track on Datasets and Benchmarks. Do not distribute.

### A.2  GH114

#### A.2.1  Details and access

The GH114 dataset was extracted from the WO2019228448 patent [3] filed by Novozymes A/S, and can be accessed at `https://patentscope.wipo.int/search/en/detail.jsf?docId=WO2019228448`, or alternatively at `https://patents.google.com/patent/WO2019228448A1/en`. The assay values/protein pairs can be found in `Table 1` in the main text (columns `SEQ ID` and `Absorbance at 405 nm - blank`) while the corresponding sequences can be found in the `Sequence Listing` document. Each protein sequence is encapsulated by `<210>`, where the number following `<210>` corresponds to a `SEQ ID` entry from patent `Table 1`. E.g., the sequence for protein `SEQ ID 12` is found between `<210> 12` and the next `<210>`. Each amino acid is described using 3-letter symbols (e.g., `Ala` for alanine). These have been processed into 1-letter symbols, and subsequently into the full sequence strings, which are collected in `data/raw/gh114/gh114.fasta`.

#### A.2.2  MSA

To strengthen the MSA, additional members from the GH114 family (PF03537) were added using the UniProt and InterPro databases [4, 5], where the sequence lengths of the added members were limited to 550 to limit the size of the final alignment, resulting in a sequence pool of 6507 sequences. The sequences are aligned using FAMSA [6].

#### A.2.3  Stratification threshold

During the dataset splitting procedure, the sequences were assigned a binary label for partition stratification. To achieve this, a two-component Gaussian mixture model was fitted to the data and used to assign labels. This corresponded to a decision boundary of 0.853.

#### A.2.4  Permission

While the data is publicly available, explicit permission to use the data for benchmarking purposes has been given by the patent's inventors, one of which is a coauthor of this paper.

### A.3  CM

#### A.3.1  Dataset details and access

The CM dataset was extracted from the supplementary materials of [7] which can be accessed at `https://www.science.org/doi/full/10.1126/science.aba3304`.

The 2133 sequences used in this paper are composed of

- 1130 naturally occurring enzymes,
- 493 bmDCA designed sequences at temperature $T = 0.33$,
- and 510 bmDCA designed sequences at temperature $T = 0.66$.

The designed sequences are obtained by Monte Carlo sampling via Boltzmann-machine learning direct coupling analysis (bmDCA) [8] and match the empirical first-, second-, and higher-order statistics of the natural homologs. The sequences also exhibit comparable catalytic levels when experimentally synthesized (see [7], Fig. 3). Given the similarity to the natural homologs in both sequence and expression, the sequences have been included.

The sequences sampled at higher temperatures (i.e., with temperature $T = 1$) and sequences designed using a simple profile model (where amino acids were only sampled according to position-specific conservation, i.e., first-order statistics) were discarded. The high-temperature sequences were almost exclusively non-functional while also being too distant from the wildtype homologs. The mean sequence identity to each sequence's nearest natural homolog was 0.55. For comparison, the mean sequence identity to nearest natural homologs for the sampled sequences at temperatures 0.33 and

0.66, is 0.81 and 0.76, respectively. While the sequences sampled using the profile model were similar in first-order statistics by design (mean sequence identity of 0.76 to nearest homologs), the sequences were exclusively non-functional. These would furthermore have been filtered out at a later stage, since only sequences with values greater than 0.42 were included in the benchmark, corresponding to high activity enzymes.

The used natural sequences are found in `aba3304_table_s1.xlsx` while the designed sequences are found in `aba3304_table_s2.xlsx`. The sequences are found aligned in the `Sequence` columns. These were stripped of the - token. The target values are found in the `norm r.e.` columns and correponds to the normalized activity relative to Escherichia coli. The proteins were named using the `No.` column while appending `seq_id_`.

### A.3.2 MSA

To strengthen the MSA, additional members from the chorismate mutase family (IPR036979) were added using the UniProt and InterPro databases [4, 5], where the sequence lengths of the added members were limited to 600 to limit the size of the final alignment, resulting in a sequence pool of 49017 sequences. The sequences are aligned using FAMSA [6].

### A.3.3 Stratification threshold

During the dataset splitting procedure, the sequences were assigned a binary label for partition stratification. To achieve this, a two-component Gaussian mixture model was fitted to the data and used to assign labels. This corresponded to a decision boundary of 0.767.

For the ablation study in which regression was performed on both active and inactive sequences, the sequences were assigned a 0 if the enzymatic activity was less than or equal to 0.42, corresponding to inactive enzymes, and a 1 if the activity was above. See [7] for details on the choice of decision boundary.

### A.3.4 Permission

While the data is publicly available, explicit consent to use the data for benchmarking purposes has been given by the authors.

### A.4 PPAT

### A.4.1 Dataset details and access

The PPAT dataset was extracted from [9] and can be accessed at `https://www.science.org/doi/10.1126/science.aao5167`. The dataset file can be found in the supplementary materials in the `aao5167_plesa-sm-tables-s8-s14.xlsx` file, sheet name `S12_PPATdata`. The sequences and target values are in the `seq` and `globalfit14` columns, respectively.

### A.4.2 MSA

To strengthen the MSA, additional members from the phosphopantetheine adenylyltransferase family (IPR001980) were added using the UniProt and InterPro databases [4, 5], where the sequence lengths of the added members were limited to 200 to limit the size of the final alignment, resulting in a sequence pool of 17891 sequences. The sequences are aligned using FAMSA [6].

### A.4.3 Stratification threshold

During the dataset splitting procedure, the sequences were assigned a binary label for partition stratification. To achieve this, a two-component Gaussian mixture model was fitted to the data and used to assign labels. This corresponded to a decision boundary of $-0.081$.

### A.4.4 Permission

While the data is publicly available, consent to use the data for benchmarking purposes was given by authors of [9].

## B   Reproducibility

All results can be reproduced using the provided shell scripts in the `scripts` directory in the code repository. A description of this process can be found in the repository's README.

Reproducing the main results (i.e., running the regression benchmark given the representations) is cheap and can be achieved in a few hours using multithreading by running the shell script `scripts/reproduce.sh`. The figures and tables can then be generated via `scripts/process_results.sh`. Generating structures and representations is more time consuming, and will be system specific. For further details, see Section B.1. We provide all used representations via the data link in Section A.1. The representations can be downloaded either in bulk with `representations.tar.gz` or individually via the `representations` directory.

All data (raw and curated) can be collected from the links provided in Section A.1. The data can be downloaded in bulk via `data.tar.gz` or individual files can be chosen through the file manager and the `data` directory.

Minor preprocessing (e.g., removing headers to make the Excel-files conform to a tabulated format) might be required before the compilation scripts in `src/data/` can be run. These preprocessed files can be found in the following files in the data repository (see Section A):

- GH114: `data/raw/gh114/gh114.csv`

- CM: `data/raw/cm/cm.csv`

- PPAT: `data/raw/ppat/ppat.xlsx`

Each dataset can then be compiled (i.e., processed and split according to the prescribed dataset splitting procedure) using `src/data/compile_<dataset>.py`. This yields the format described in Section A.

The final partitioning as determined using GraphPart [10] is dependent on the ordering of the input data. Shuffling the datasets, i.e., changing the order of the sequences, will thus slightly change the partitions. We observed only minor changes to the benchmark results given these slight differences.

The CT, ESM-1B, ESM-2, ESM-IF1, MIF-ST, MSA (1-HOT) as well as ESM-IF1 likelihoods can be generated using the `generate_representations.sh` script.

The Evoformer embeddings are extracted during folding using AlphaFold2 by using the `--save-single-representations` flag of ColabFold [2].

To generate the EVE embeddings, the model has to be trained. This can be handled via the `train_EVE_models.sh` script. EVE is trained on each dataset a total of three times using different seeds. The ELBO scores and embeddings are computed/extracted from each trained model. The embeddings are placed in the `0/1/2` subdirectories of `representations/<dataset>/EVE/`.

### B.1   Computational resources

A system with an Intel Xeon E5-2680v4 CPU, NVIDIA RTX A5000 GPUs, and 512 GB of RAM was used for benchmarking, computing ESM/MIF-ST embeddings, and training EVE models (though the benchmarking process itself does not utilize GPUs). A system with an AMD EPYC 7642 CPU, NVIDIA A40 GPUs, and 1 TB of RAM was used for protein folding.

A conservative estimate puts the computational resources for each sequence at 4 minutes, which for 2804 sequences results in approximately 187 GPU hours. The majority of this time (>80 %) is spent

 folding the proteins using AlphaFold2. Running the regression benchmark takes approximately 3
 hours using a multithreading-capable CPU.

 # C   Dataset target histograms

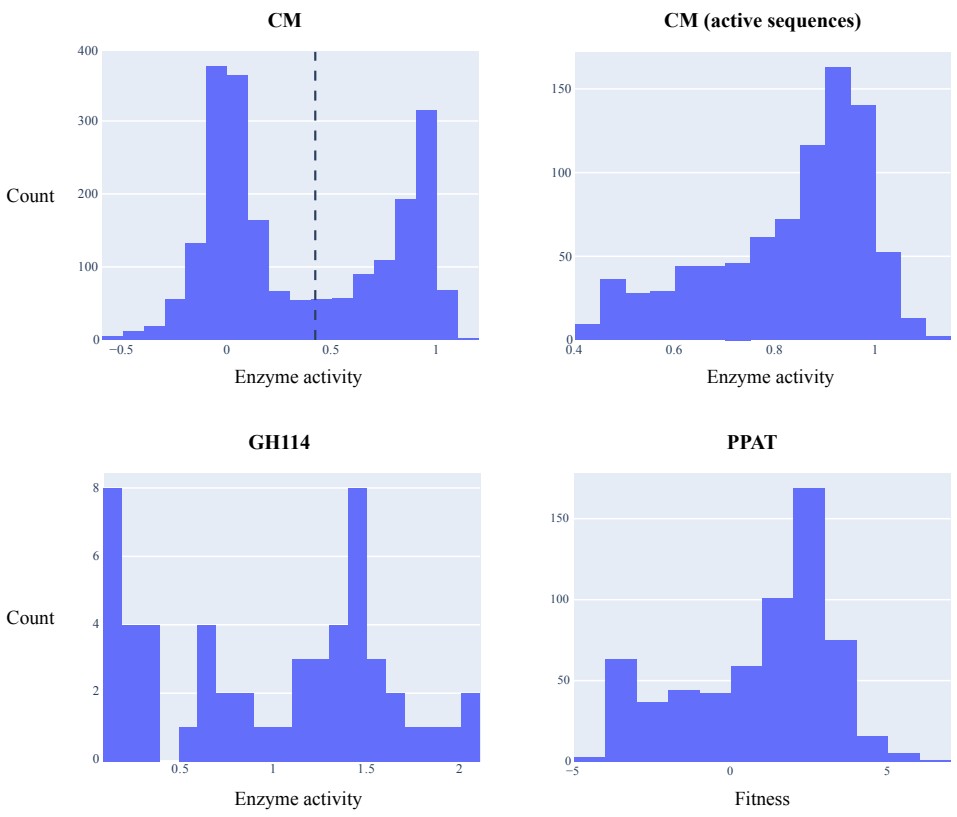

Figure A1: Target histograms of the datasets. CM dataset shows both full dataset prior to filtering and
the subset of active sequences that is included in the benchmark. The subset includes only sequences
with enzyme activities $> 0.42$.

# D Histograms of cross-validation partitions

## D.1 GH114

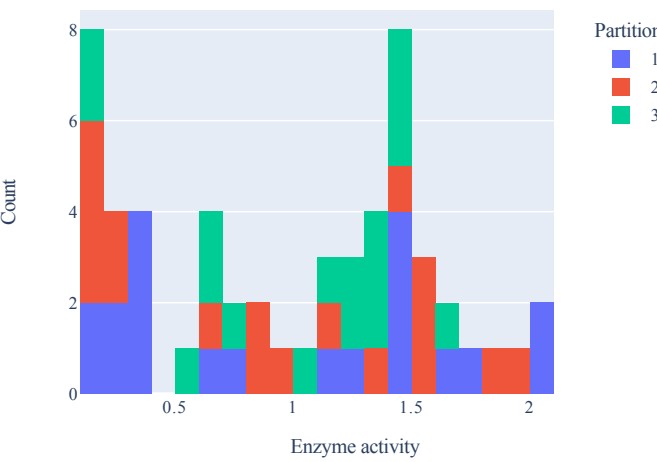

Figure A2: Stacked histogram over distribution of target values for GH114 dataset. Each color correspond to a partition.

## D.2 CM

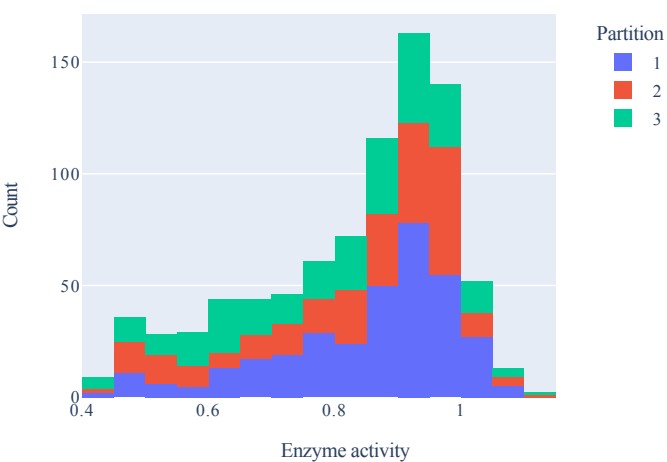

Figure A3: Stacked histogram over distribution of target values for CM dataset. Each color correspond to a partition.

 **D.3 PPAT**

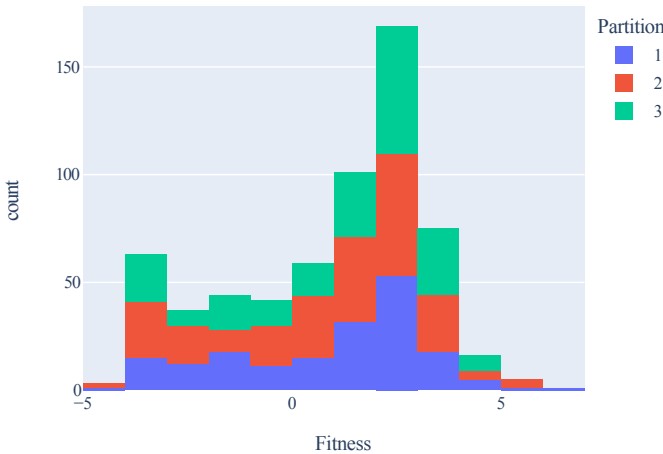

Figure A4: Stacked histogram over distribution of target values for PPAT dataset. Each color correspond to a partition.

## E  Phylogenetic trees for PPAT dataset

The phylogenetic tree in Figure 2 was constructed based on a family-wide multiple sequence alignment using FastTree [11]. The extracted segment corresponds to the top right quarter.

### E.1  Phylogenetic tree colored by dataset partitioning scheme

The phylogenetic tree in Figure A5 is the full version of the leftmost segment in Figure 2.

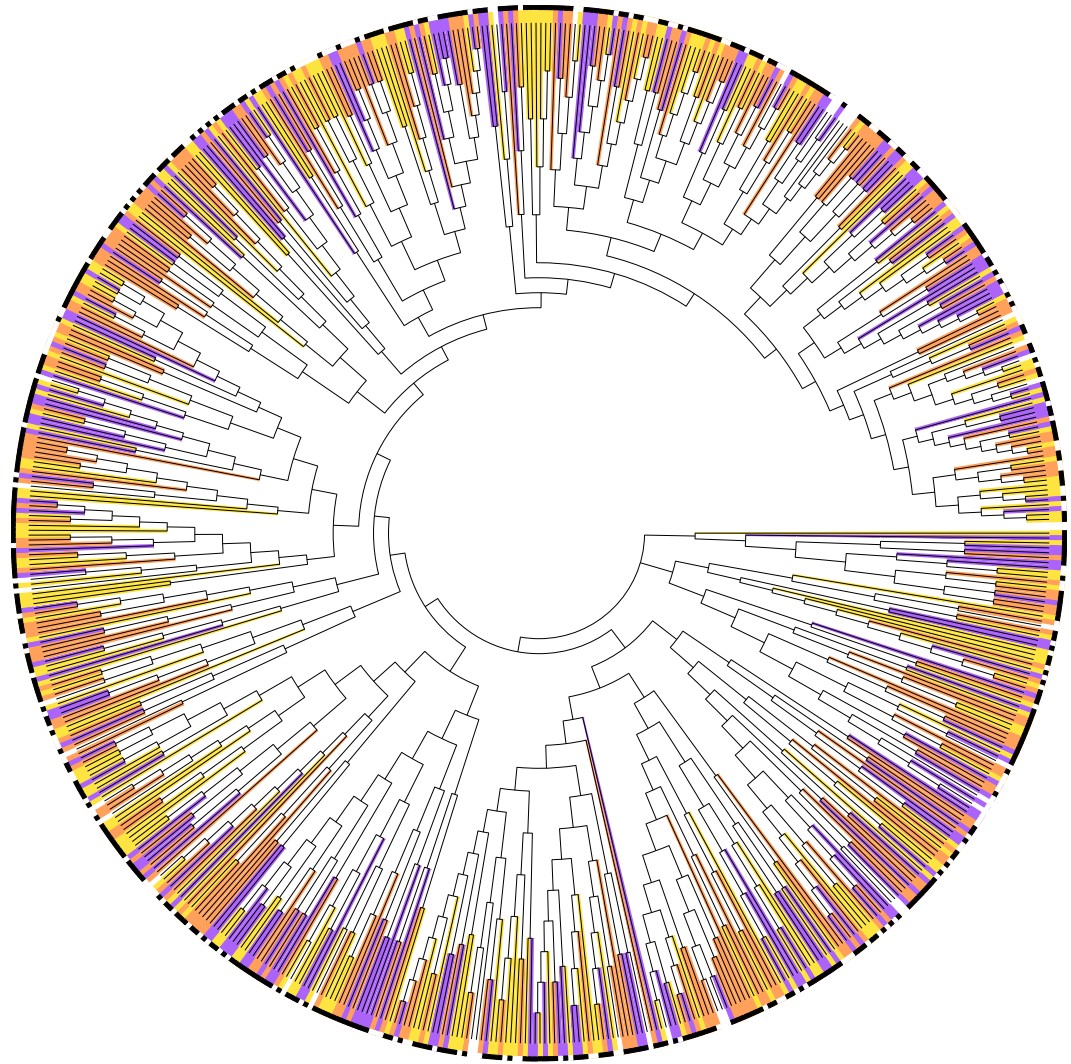

Figure A5: Phylogenetic tree for PPAT dataset. Each sequence is colored according to its partition as computed in the data splitting setup. Black squares indicate high target value while white squares indicate low target value.

## E.2 Phylogenetic tree colored by MMseqs-based clustering scheme

The phylogenetic tree in Figure A6 is the same tree as in Figure A5 with a different coloring scheme. The protein sequences were clustered using MMseqs [12] such that at least two large clusters were created. These two large clusters get separate colors, while the remaining minor clusters get a shared color. This represents an alternative dataset splitting scheme. As is apparent from the figure, wide bands of uniformly colored (and thus partitioned) sequences appear. Large subfamilies are all placed in the same partition which means that learning across subfamilies is difficult. The partitioning is furthermore not stratified which might result in low-scoring partitions.

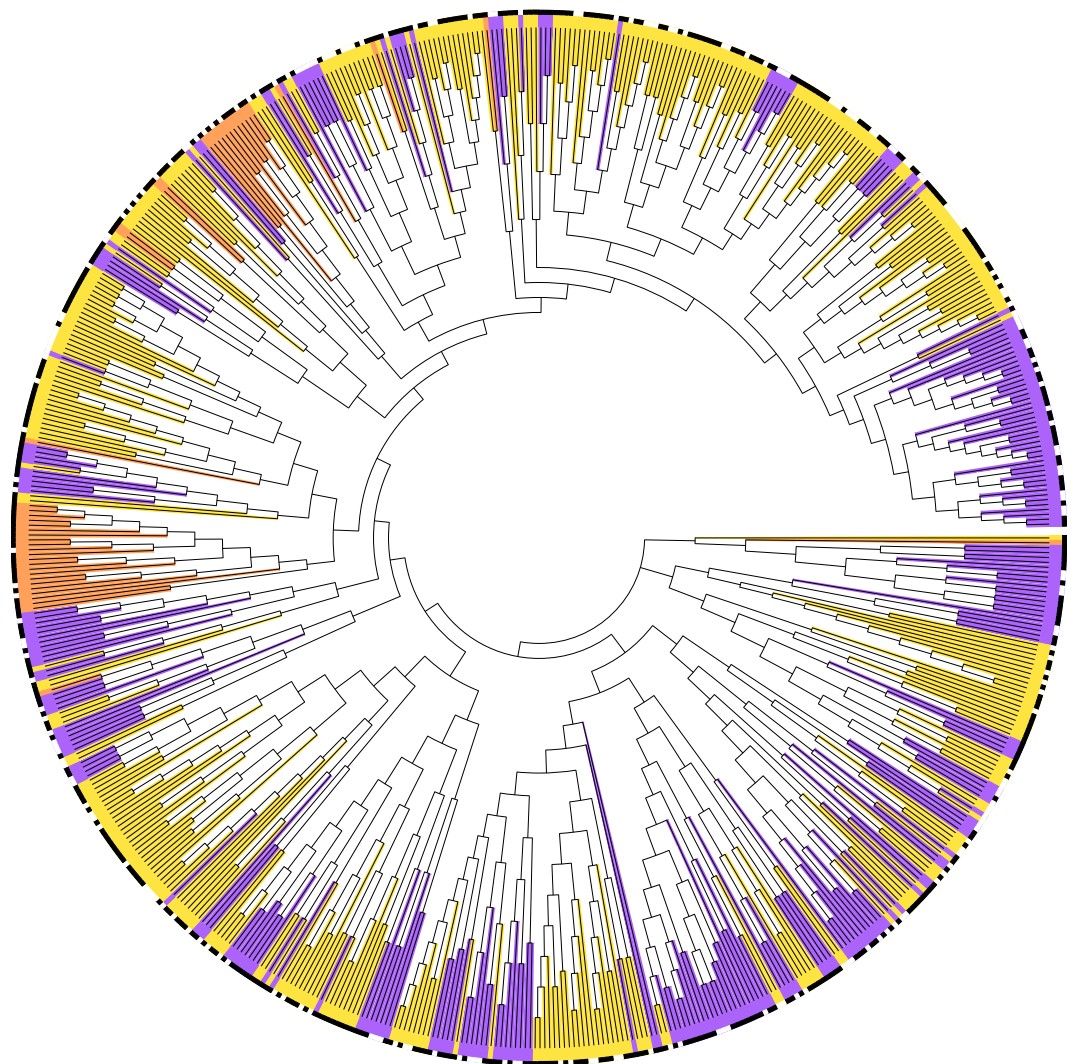

Figure A6: Phylogenetic tree for PPAT dataset. Each sequence is colored according to its partition as computed in the data splitting setup. Black squares indicate high target value while white squares indicate low target value.

## E.3 Phylogenetic tree colored randomly

The phylogenetic tree in Figure A7 is the same tree as in Figure A5 with a different coloring scheme. Instead of relying on the prescribed partitioning strategy, each sequence is assigned one of the three colors randomly. This corresponds to generating three random partitions. While the tree looks similar to the one in Figure A5, there is no guarantee that nearly identical sequences are not placed in separate partitions thus allowing for data leakage. There is furthermore no mechanism to ensure properly stratified splits (although this can be handled in most machine learning frameworks).

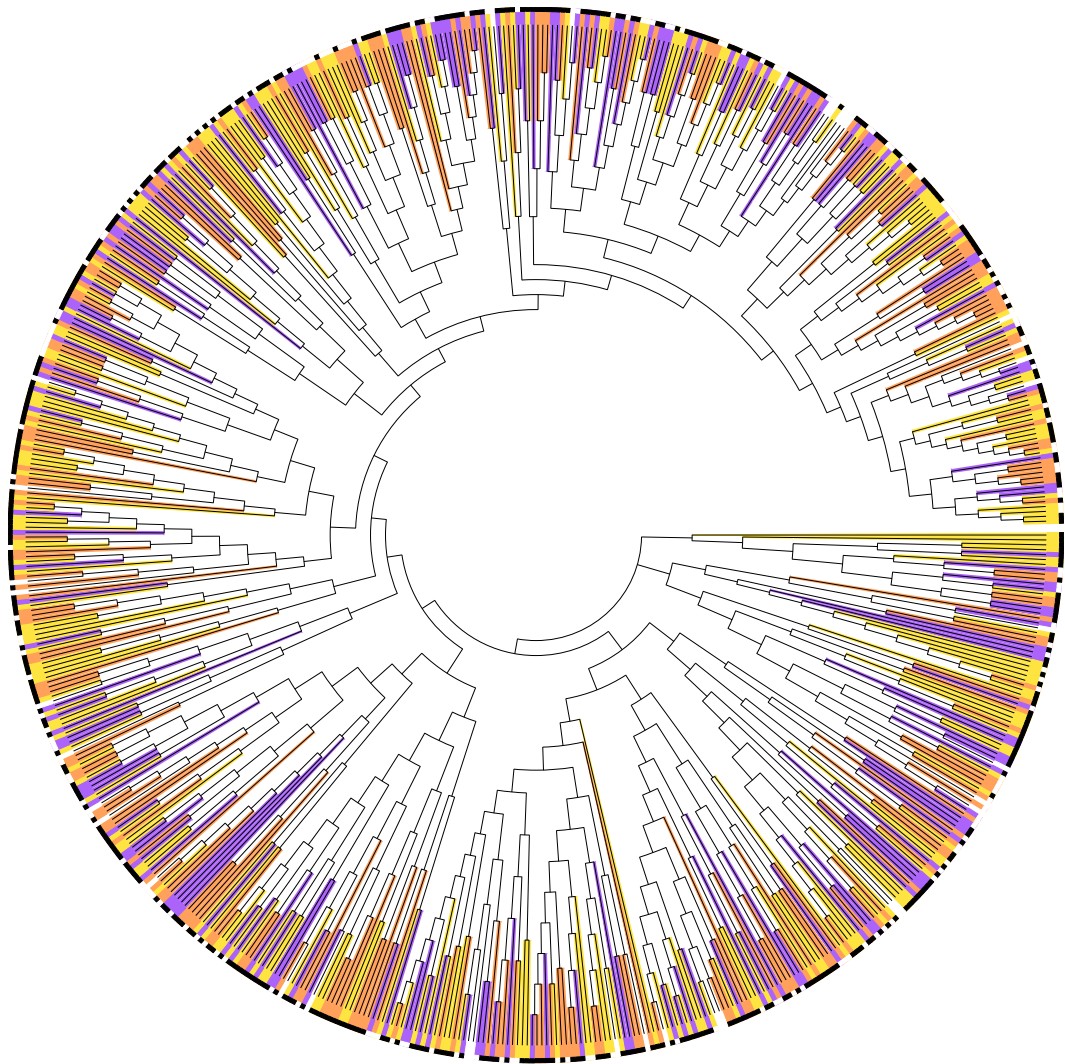

Figure A7: Phylogenetic tree for PPAT dataset. Each sequence is colored randomly corresponding to a random splitting procedure. Black squares indicate high target value while white squares indicate low target value.

# F  ProteinGym sequence identities

Table A1 shows the median, mean. and standard deviation of the pairwise sequence identities for each benchmark dataset. For comparison, we have computed the same quantities for 48 substitution tasks present in the ProteinGym [13] set of deep mutational scanning assays which is commonly used for benchmarking variant effect predictors. These quantities can be seen in Table A2. The stark differences shows the diversity of the wildtype datasets.

Table A1: Diversity of FLOP datasets

| Dataset | Median %ID | Mean %ID | Standard deviation |
|---|---|---|---|
| GH114 | 0.485 | 0.514 | 0.098 |
| CM | 0.400 | 0.408 | 0.059 |
| PPAT | 0.513 | 0.515 | 0.046 |
| **Mean** | **0.466** | **0.479** | **0.067** |

Table A2: Diversity of ProteinGym datasets

| Dataset | Median %ID | Mean %ID | Standard deviation |
|---|---|---|---|
| A0A140D2T1_ZIKV_Sourisseau_growth_2019 | 0.999 | 0.999 | 0.000 |
| A0A192B1T2_9HIV1_Haddox_2018 | 0.998 | 0.998 | 0.000 |
| A0A2Z5U3Z0_9INFA_Doud_2016 | 0.996 | 0.997 | 0.001 |
| A0A2Z5U3Z0_9INFA_Wu_2014 | 0.996 | 0.996 | 0.000 |
| A4GRB6_PSEAI_Chen_2020 | 0.992 | 0.993 | 0.001 |
| AMIE_PSEAE_Wrenbeck_2017 | 0.994 | 0.995 | 0.001 |
| B3VI55_LIPST_Klesmith_2015 | 0.995 | 0.996 | 0.001 |
| BLAT_ECOLX_Deng_2012 | 0.993 | 0.993 | 0.001 |
| BLAT_ECOLX_Firnberg_2014 | 0.993 | 0.993 | 0.001 |
| BLAT_ECOLX_Jacquier_2013 | 0.993 | 0.993 | 0.000 |
| BLAT_ECOLX_Stiffler_2015 | 0.993 | 0.993 | 0.000 |
| BRCA1_HUMAN_Findlay_2018 | 0.999 | 0.999 | 0.000 |
| C6KNH7_9INFA_Lee_2018 | 0.996 | 0.997 | 0.001 |
| CALM1_HUMAN_Weile_2017 | 0.987 | 0.987 | 0.002 |
| CCDB_ECOLI_Adkar_2012 | 0.980 | 0.980 | 0.001 |
| CCDB_ECOLI_Tripathi_2016 | 0.980 | 0.980 | 0.001 |
| DLG4_RAT_McLaughlin_2012 | 0.997 | 0.997 | 0.000 |
| ENV_HV1B9_DuenasDecamp_2016 | 0.998 | 0.998 | 0.000 |
| ENV_HV1BR_Haddox_2016 | 0.998 | 0.998 | 0.000 |
| GAL4_YEAST_Kitzman_2015 | 0.998 | 0.998 | 0.000 |
| HSP82_YEAST_Flynn_2019 | 0.997 | 0.997 | 0.000 |
| HSP82_YEAST_Mishra_2016 | 0.997 | 0.997 | 0.000 |
| I6TAH8_I68A0_Doud_2015 | 0.996 | 0.996 | 0.000 |
| IF1_ECOLI_Kelsic_2016 | 0.972 | 0.974 | 0.005 |
| KKA2_KLEPN_Melnikov_2014 | 0.992 | 0.993 | 0.001 |
| MK01_HUMAN_Brenan_2016 | 0.994 | 0.994 | 0.000 |
| MTH3_HAEAE_Rockah-Shmuel_2015 | 0.994 | 0.994 | 0.000 |
| NCAP_I34A1_Doud_2015 | 0.996 | 0.996 | 0.000 |
| P84126_THETH_Chan_2017 | 0.992 | 0.992 | 0.000 |
| PA_I34A1_Wu_2015 | 0.997 | 0.998 | 0.001 |
| POLG_CXB3N_Mattenberger_2021 | 0.999 | 0.999 | 0.000 |
| POLG_HCVJF_Qi_2014 | 0.999 | 0.999 | 0.000 |
| PTEN_HUMAN_Mighell_2018 | 0.995 | 0.995 | 0.000 |
| Q2N0S5_9HIV1_Haddox_2018 | 0.998 | 0.998 | 0.000 |
| Q59976_STRSQ_Romero_2015 | 0.996 | 0.997 | 0.001 |
| RASH_HUMAN_Bandaru_2017 | 0.989 | 0.989 | 0.000 |
| REV_HV1H2_Fernandes_2016 | 0.983 | 0.983 | 0.001 |
| RL401_YEAST_Mavor_2016 | 0.984 | 0.985 | 0.003 |
| RL401_YEAST_Roscoe_2013 | 0.984 | 0.985 | 0.002 |
| RL401_YEAST_Roscoe_2014 | 0.984 | 0.985 | 0.002 |
| SC6A4_HUMAN_Young_2021 | 0.997 | 0.997 | 0.000 |
| SUMO1_HUMAN_Weile_2017 | 0.980 | 0.981 | 0.003 |
| TAT_HV1BR_Fernandes_2016 | 0.977 | 0.977 | 0.001 |
| TPK1_HUMAN_Weile_2017 | 0.992 | 0.992 | 0.001 |
| TRPC_SACS2_Chan_2017 | 0.992 | 0.992 | 0.000 |
| TRPC_THEMA_Chan_2017 | 0.992 | 0.992 | 0.000 |
| UBC9_HUMAN_Weile_2017 | 0.987 | 0.988 | 0.002 |
| UBE4B_MOUSE_Starita_2013 | 0.998 | 0.998 | 0.000 |
| **Mean** | **0.993** | **0.992** | **0.001** |

## G   Representation dimensionalities

The dimensionalities of the different protein representations are shown in Table A3. The ESM, Evoformer, and MIF-ST embeddings are mean-pooled along the protein length dimension to obtain fixed inputs.

A multiple sequence alignment (MSA) is generated for each (enriched) protein family, resulting in different dimensionalities. The amino acids are then one-hot encoded to a `MSA_length` $\times 20$ matrix for each protein, which is in turn flattened to a vector input.

The CT representation consists of two parts: *compositional* and *transitional* descriptors which are concatenated. Each of the two groups in turn consists of seven physicochemical descriptors, relating to overall polarizability, charge, hydrophobicity, polarity, secondary structure, solvent accessibility, and van der Waals volume of a sequence. Each descriptor is in turn represented by three numbers. This yields a total of $2 \times 7 \times 3 = 42$ dimensions. For descriptions of the various features, see `https://github.com/gadsbyfly/PyBioMed/blob/45440d8a70b2aa2818762ceadb499dd3a1df90bc/PyBioMed/PyProtein/CTD.py#L60` and [14].

Table A3: Dimensionalities of the different protein representations.

| Representation | D | Note | Model name |
|---|---|---|---|
| CT | 42 | – | – |
| ESM-1B | 1280 | Mean-pooled | `esm1b_t33_650M_UR50S` |
| ESM-2 | 2560 | Mean-pooled | `esm2_t36_3B_UR50D` |
| ESM-IF1 | 256 | Mean-pooled | `esm_if1_gvp4_t16_142M_UR50` |
| MIF-ST | 256 | Mean-pooled | `mifst` |
| EVE | 50 | Seeds `0,1,2` | – |
| Evoformer (AF2) | 256 | Mean-pooled | `alphafold2_multimer_v3` |
| MSA (1-HOT, GH114) | 88420 | Flattened | 6507 sequences in MSA. |
| MSA (1-HOT, CM) | 109980 | Flattened | 49017 sequences in MSA. |
| MSA (1-HOT, PPAT) | 10140 | Flattened | 17891 sequences in MSA. |

## H   EVE

Due to the stochastic training process, we train EVE on each fitness landscape using three different random seeds (`0,1,2`). The reported performance will thus be the average over the predictions using the three different representations for each sequence. While EVE was originally used to predict variant effects of single wildtype proteins, it can be used on any multiple sequence alignment. The built-in preprocessing requires a reference wildtype (query) sequence. This query sequence is then used to trim and otherwise clean the remaining sequences in the MSA. Since no single wildtype is representative for entire protein families, we instead generate an artificial query sequence. Given the full-length MSA, we iterate through all of our labelled sequences (a minor part of the full MSA), and create a query sequence which has an amino acid (we arbitrarily chose 'A') at any position in the MSA, where any of the labelled sequences also have an amino acid. The remaining positions are filled with gaps. For example, say that sequences `-A-T-H` and `-AT-J-` are two labelled sequences from the MSA. The corresponding query sequence would thus be `-AA-AA`. The query sequence is only used in the preprocessing, e.g., to conserve the columns, where the labelled sequences have occupancy, and to remove columns where none do. The query sequence is not included in the model training itself. Alternative preprocessing is equally viable which can avoid the creation of the artificial query sequence.

## I  ProteinMPNN

ProteinMPNN [15] is an inverse folding model. As described in example 3 in the repository, the model can estimate its uncertainty given structure/sequence pairs by using the `score_only` functionality. We use the `v_48_020` weights, sampling temperature of 0.1, and number of sequences per target of 5.

## J  Tranception

We evaluate the fitness of the wildtype sequences using the bidirectional scoring with retrieval using the Tranception L (Large) as defined in the manuscript [13]. This utilises a multiple sequence alignment for each sequence during scoring.

## K  Regressor hyperparameters

In each cross validation iteration, the regressor is optimized via a grid search. The regressor is trained with all configurations on the training set, and the model providing the lowest mean squared error on the validation set is used to predict on the test set. In addition to the shown results from a random forest regressor, the results from K-nearest neighbour model, a ridge regressor, and a multilayer perceptron (MLP) are also computed. The following hyperparameter grids are used:

- `Ridge(random_state=0)`: Regularization strength was chosen among: 0.0001, 0.001, 0.01, 0.1, 0.2, 0.5, 1, 2, 10, 25, 50, 100.
- `KNeigborsRegressor()`: The number of neighbours was chosen among: 1, 2, 5, 10, 25. For the GH114 dataset, the 10 and 25 options were removed due to the small partition sizes.
- `RandomForestRegressor(random_state=0)`: Minimum samples to split was chosen among 2, 5. Maximum number of features was either `sqrt` or `log2`. Number of estimators was either 100 or 200.
- `MLPRegressor(random_state=0, max_iter=2000)`: hidden layer sizes was either 10 or 100, the L2 regularization strength was set to 0, 0.01, or 0.0001, while the optimizer was either Adam (with gradient descent) or L-BFGS.

We use the scikit-learn implementations of the regressors [16]. The parameters not explicitly defined above are the default parameters. Several other grids for the four models were examined but provided no significant performance increases. The MLP-regressor occasionally experienced convergence issues (with both optimizers).

## L  Ablation results figure

The values in Table 3 are shown as bar plots in Figure A8. The figure has been moved to the appendix due to page limit constraints.

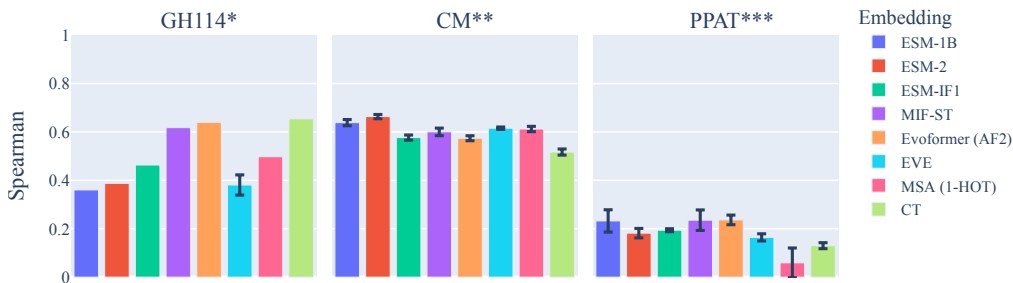

Figure A8: Spearman's correlation coefficient between predictions and targets over test partitions, grouped by dataset. Standard error is shown as vertical bars. *: Hold-out validation. **: Regression on both active and inactive proteins. ***: Repeated random splitting.

## L.1 Hold-out ablation study on all datasets

The included ablation study shows the results if hold-out validation is applied to the GH114 dataset

using a ridge regressor. In Figure A9 is shown the same ablation study on all three datasets using

a K-nearest neighbour regressor, a ridge regressor, and a random forest regressor. For EVE, three
models have been trained at different initializations thereby explaining the errors bars.

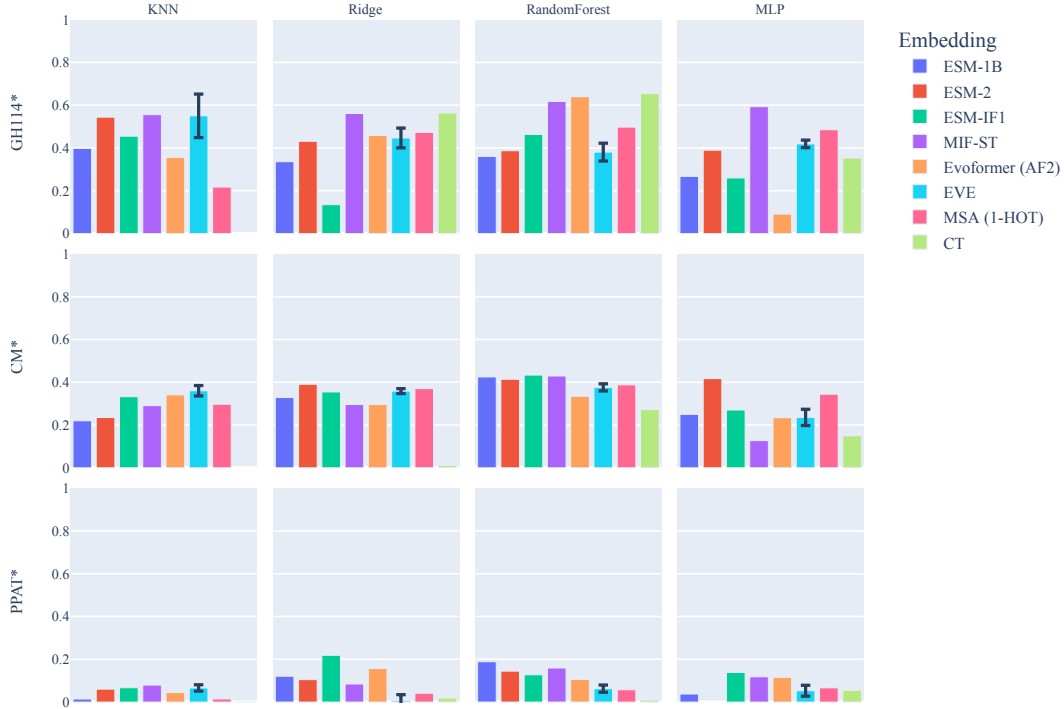

Figure A9: Spearman's rank correlation coefficient between predictions and targets using a hold-out
validation approach, grouped by regressor and dataset.

 **L.2   Random splitting ablation study on all datasets**

253  The included ablation study shows the results if splitting is applied to the PPAT dataset using a ridge
254  regressor. In Figure A10 is shown the same ablation study on all three datasets using a K-nearest
neighbour regressor, a ridge regressor, and a random forest regressor.

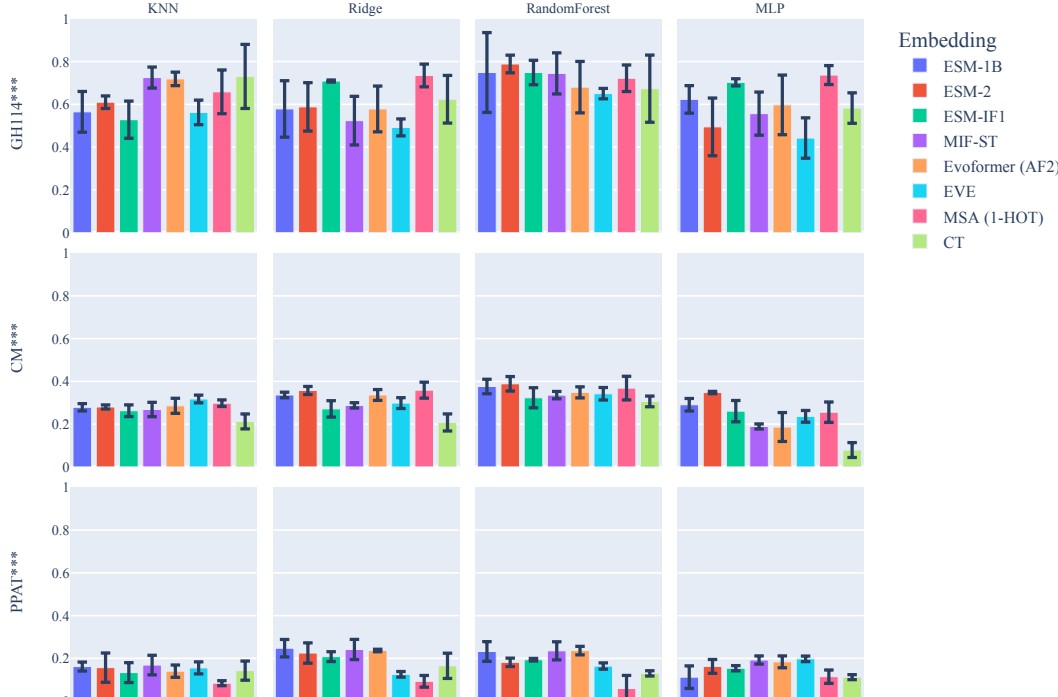

Figure A10: Spearman's rank correlation coefficient between predictions and targets over using a
cross-validation approach with randomly sampled partitions repeated on on three random seeds,
255  grouped by regressor and dataset.

## M    Classification results for CM dataset

Classification was carried out on a combined pool of inactive and active sequences for the CM dataset. The threshold between the two classes is set to 0.42 as described in [7]. The procedure was carried out just as described in Section 3.1 simply with alternative targets and objectives. The results using a K-nearest neighbour classifier, a logistic regression classifier, a random forest classifier, and a multi-layer perceptron are shown in Figure A11. The models were optimized using a binary cross-entropy loss function. The shown metric is Matthew's correlation coefficient. As can be seen from the results, the classification task is significantly easier than the proposed regression benchmark. This supports the notion of carrying out an initial classification prior to performing regression on the subset of active sequences.

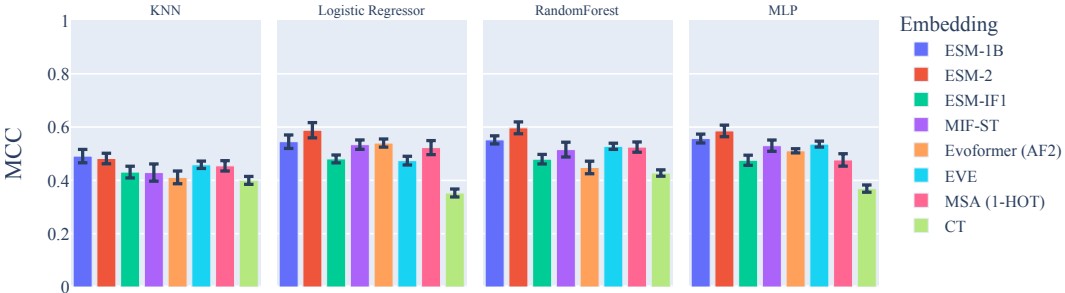

Figure A11: Average Matthew's correlation coefficient between predictions and targets over test partitions. Standard error is shown as vertical bars.

# N    Additional results

## N.1    Results using additional regressors (Spearman)

Test results obtained using a K-nearest neighbour regressor, a ridge regressor (as shown in the main text), a random forest regressor, and an MLP are shown in Figure A12. We observe no systematic differences between the choice of regressor, other than the random forest consistently reaching high performance. This led us to include only the results from the random forest predictor in the main text.

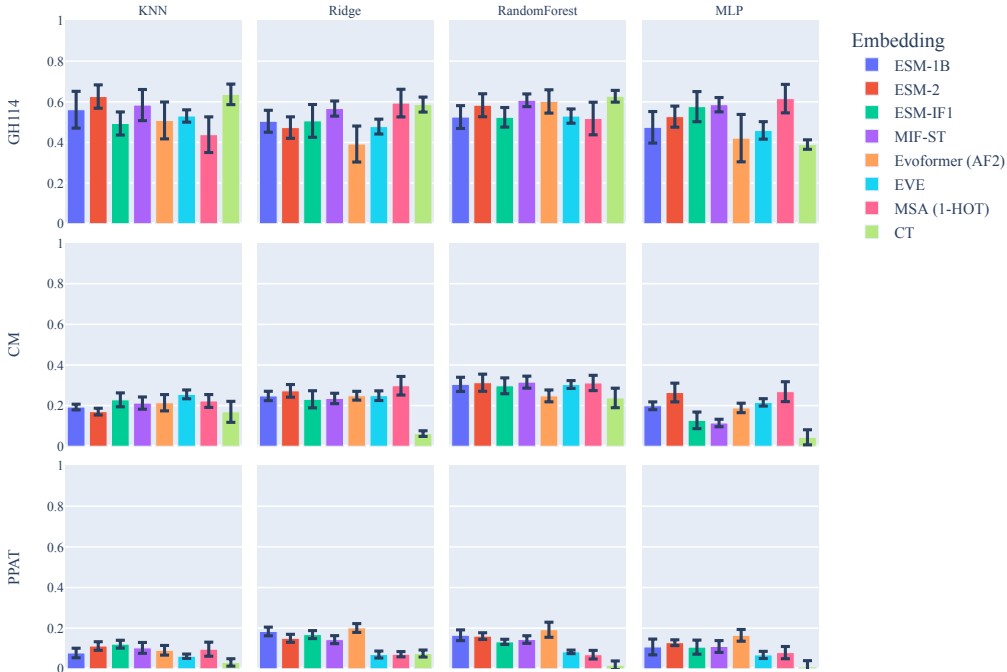

Figure A12: Average Spearman's correlation between predictions and targets over test partitions, grouped by regressor and dataset. Standard error is shown as vertical bars.

## N.2 Benchmark results (RMSE)

Test RMSE obtained can be seen in Table A4.

Table A4: Benchmark results with random forest regressor. Mean RMSE and standard error using cross-validation. Lower is better.

|  | GH114 | CM | PPAT |
|---|---|---|---|
| ESM-1B | **0.43** $\pm$ 0.04 | **0.15** $\pm$ 0.0 | **2.32** $\pm$ 0.03 |
| ESM-2 | **0.43** $\pm$ 0.04 | **0.15** $\pm$ 0.0 | **2.33** $\pm$ 0.03 |
| ESM-IF1 | 0.48 $\pm$ 0.04 | **0.15** $\pm$ 0.0 | **2.33** $\pm$ 0.02 |
| MIF-ST | **0.42** $\pm$ 0.04 | **0.15** $\pm$ 0.0 | **2.34** $\pm$ 0.03 |
| Evoformer (AF2) | 0.45 $\pm$ 0.05 | 0.16 $\pm$ 0.0 | **2.32** $\pm$ 0.03 |
| EVE | **0.44** $\pm$ 0.02 | 0.16 $\pm$ 0.0 | 2.41 $\pm$ 0.01 |
| MSA (1-HOT) | **0.45** $\pm$ 0.04 | **0.15** $\pm$ 0.0 | **2.35** $\pm$ 0.02 |
| CT | **0.45** $\pm$ 0.05 | 0.16 $\pm$ 0.0 | 2.41 $\pm$ 0.03 |

## N.3 Results using additional regressors (RMSE)

Test RMSE obtained using a K-nearest neighbour regressor, a ridge regressor (as shown in the main text), a random forest regressor, and an MLP are shown in Figure A13. Note that the y-axes are not shared.

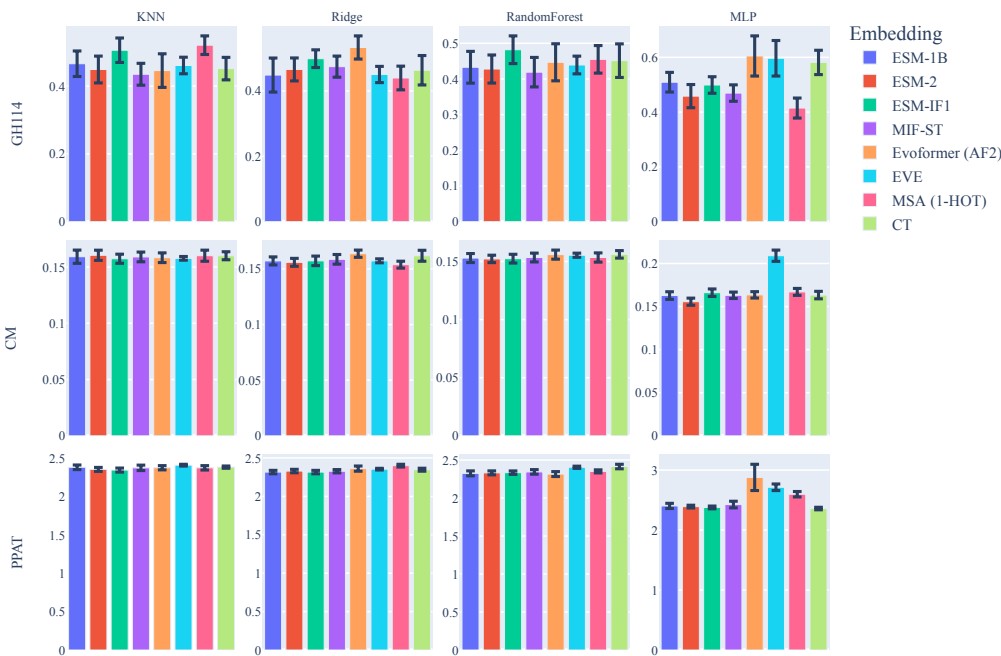

Figure A13: Average RMSE over test partitions, grouped by regressor and dataset. Standard error is shown as vertical bars.

## N.4 Results for CM dataset when using only natural homologs

During the curation process of the chorismate mutase dataset, the 1130 natural homologs were enriched with 1003 model-generated sequences (for details, see Appendix A.3. The benchmark results if only the natural sequences were used can be seen in Figure A14.

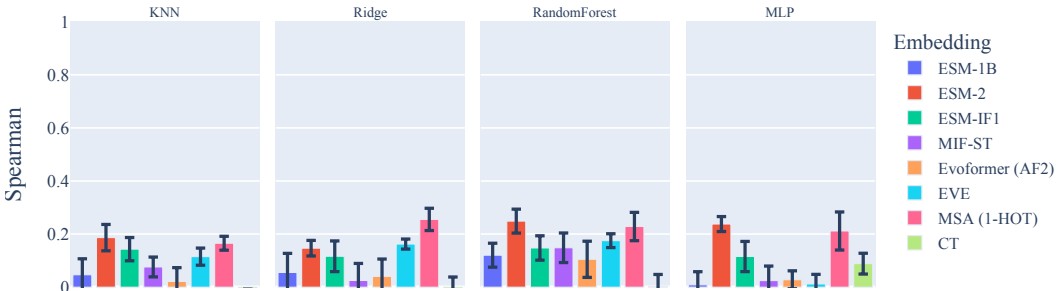

Figure A14: Average Spearman correlation coefficient between predictions and targets over test partitions. Standard error is shown as vertical bars.

## O Retraction from ICLR 2022

A previous version of this work was submitted to – and subsequently withdrawn from – the *International Conference on Learning Representations* (ICLR) 2022. The earlier version had a lack of novelty and limited relevance. The paper has seen major revisions since, including removing an earlier dataset, introducing the GH114 dataset, a more elaborate description of the limitations of previous work with respect to wildtype exploration, a more thorough description of the methodology and its impact, thorough supplementary materials and more.

# P   Mandatory dataset information details

All curated datasets are publicly available with thorough documentation (see Section A) and consent to use the three datasets for benchmarking purposes has been given by the respective authors. Since the GH114 dataset has not been used in the literature prior to our work, however, we here include the mandatory details – where/if relevant – for new datasets. Headings are in italics and answers are in default format.

1. *Submission introducing new datasets must include the following in the supplementary materials*:

   (a) *Dataset documentation and intended uses. Recommended documentation frameworks include datasheets for datasets, dataset nutrition labels, data statements for NLP, and accountability frameworks.*

   The documentation for GH114 can be found in the main text of the patent [3] at `https://patentscope.wipo.int/search/en/detail.jsf?docId=WO2019228448`. Intended use of the data in this body of work is for benchmarking purposes, as illustrated in the main article.

   (b) *URL to website/platform where the dataset/benchmark can be viewed and downloaded by the reviewers.*

   Instructions for how to access both raw and processed/curated data can be found in Section A.1. The repository at `https://github.com/petergroth/FLOP` holds additional details for accessing remaining data and precomputed representations.

   (c) *Author statement that they bear all responsibility in case of violation of rights, etc., and confirmation of the data license.*

   All protein sequences in the GH114 dataset are patented and all rights belong to the patent holders. Consent to use the data for benchmarking purposes was given by the patent holders directly.

   (d) *Hosting, licensing, and maintenance plan. The choice of hosting platform is yours, as long as you ensure access to the data (possibly through a curated interface) and will provide the necessary maintenance.*

   All data (raw and processed) is kept in an archive managed by the *Electronic Research Data Archive* (ERDA) by the University of Copenhagen. The data can be accessed at `https://sid.erda.dk/sharelink/HLXs3e9yCu`. The raw data itself is available via the patent itself (see item (a)).

2. *To ensure accessibility, the supplementary materials for datasets must include the following:*

   (a) *Links to access the dataset and its metadata. This can be hidden upon submission if the dataset is not yet publicly available but must be added in the camera-ready version. In select cases, e.g when the data can only be released at a later date, this can be added afterward. Simulation environments should link to (open source) code repositories.*

   For links to the datasets (and code), see Section A and item (f) below.

   (b) *The dataset itself should ideally use an open and widely used data format. Provide a detailed explanation on how the dataset can be read. For simulation environments, use existing frameworks or explain how they can be used.*

   A detailed description of dataset formats and of how the dataset can be used can be found in Section A. See item (f) for links.

   (c) *Long-term preservation: It must be clear that the dataset will be available for a long time, either by uploading to a data repository or by explaining how the authors themselves will ensure this.*

   All used data (raw, processed, representations) is stored by the Electronic Research Data Archive (ERDA) by the University of Copenhagen. The curated datasets used for benchmarking can additionally be found in the GitHub repository (see item (f) for links).

(d) *Explicit license: Authors must choose a license, ideally a CC license for datasets, or an open source license for code (e.g. RL environments).*

While we do not hold the rights to the datasets, our contribution in the form of establishing the benchmark (i.e., the methodology and code) falls under the open source MIT License. As described in the supplementary materials, we ask that references to the presented tasks include references to the original sources.

(e) *Add structured metadata to a dataset's meta-data page using Web standards (like schema.org and DCAT): This allows it to be discovered and organized by anyone. If you use an existing data repository, this is often done automatically.*

No metadata was added or altered to the data and remains accessible (see item (a)).

(f) *Highly recommended: a persistent dereferenceable identifier (e.g. a DOI minted by a data repository or a prefix on identifiers.org) for datasets, or a code repository (e.g. GitHub, GitLab,...) for code. If this is not possible or useful, please explain why.*

Data access: `https://sid.erda.dk/sharelink/HLXs3e9yCu`. GitHub repository for all code and more details: `https://github.com/petergroth/FLOP`.

3. *For benchmarks, the supplementary materials must ensure that all results are easily reproducible. Where possible, use a reproducibility framework such as the ML reproducibility checklist, or otherwise guarantee that all results can be easily reproduced, i.e. all necessary datasets, code, and evaluation procedures must be accessible and documented.*

See Section B.