# OpenReview forum: "FLOP: Tasks for Fitness Landscapes Of Protein wildtypes"
_NeurIPS.cc/2023/Track/Datasets_and_Benchmarks — Submitted to NeurIPS 2023 Datasets and Benchmarks_

### Official Review · Reviewer_YyxM · 2023-07-19
**Benchmarking fitness landscapes of proteins**

**Rating:** 7
**Confidence:** 2
**Correctness:** I am not an expert in protein enginee…
**Clarity:** Yes

**Strengths:**

The paper is well written. The authors clearly differentiate their work from prior benchmarks. The use of specialized CV splits is well motivated. Several algorithms are benchmarked.

**Additional Feedback:**

None

**Documentation:**

Yes

**Limitations:**

No, see above

**Opportunities For Improvement:**

The authors do not discuss the limitations of their benchmark design, only the limitations of the model results. They could provide better detail in this regard to motivate improvements.

**Relation To Prior Work:**

Yes

**Summary And Contributions:**

The authors contribute a benchmark for predicting function of wildtype proteins for wildtype discovery. A key part of this benchmark is to address train/ test set construction in the presence of non-iid data. The authors benchmark traditional ML (random forests) and some zero shot learners on three datasets they construct and find mixed results of current models suggesting prior comparisons of model performance for protein wildtype discovery may be overly optimistic.

---

> ### Author Response · Authors · 2023-08-18
> **Rebuttal**
>
> We would like to thank the reviewer for their inputs.
>
> **Improvements**
> - *The authors do not discuss the limitations of their benchmark design, only the limitations of the model results. They could provide better detail in this regard to motivate improvements.*
>
> The reviewer raises a valid point in that we can more explicitly discuss and reflect on possible limitations with our proposed benchmark. Largely driven by the limitations raised by reviewer aTwL and this review, we have altered the conclusion by now including an extensive discussion on these potential limitations.
>
> One concern is the low number of included datasets as well as their relatively small sizes (i.e. number of included sequence/target pairs). Wildtype exploration. which we're essentially benchmarking, can be viewed as a predominantly explorative phase of protein optimization, which is then followed by the exploitation phase of subsequent protein engineering. Often, limited resources are allocated to wildtype exploration since it is inherently costly and somewhat wasteful as it tends to produce many poor candidates. This is not likely to change *unless* we find ways to improve our wildtype search strategy. This will in turn require better
> predictions. We therefore consider the limited dataset sizes as an inherent condition and limitation in this domain.
>
> Another potential limitation is impact of the benchmark. Our results show that the supervised approaches outperform the zero-shot approaches, but that no one representation/representation paradigm consistently outperforms the others. This could suggest that the employed representations are not sufficiently informative. A key limitation for a number of the included representations is that we obtained protein-level representations as averages over the protein length to arrive at fixed-length embeddings, which is known to be suboptimal. We encourage the community to experiment with novel aggregation strategies and new representation designs to improve performance on our benchmark. It is also conceivable that general-purpose protein representation models might not by themselves be sufficient to convincingly improve on the proposed tasks. One can imagine that further improvements can be obtained using pretrained models
> fine-tuned on a protein family of interest – or by developing weakly-supervised representation models incorporating relevant properties that correlate with the function of interest (e.g. thermostability).
>
> Lastly, we for 2 of the 3 tasks, observed relatively low test correlations which can raise concerns of impact. Although the performance of current baselines on some of our test-cases is fairly low in absolute terms, even low correlations can provide useful guidance on selecting wildtype protein starting points and can have measurable real-world impacts since it is a process which is used on a daily basis.
>
> We thank the reviewer for pointing out the potential to discuss limitations to a greater extent which we - with the additional reviews - believe has led to an improved article.

---

> > ### Comment · Reviewer_YyxM · 2023-08-29
> >
> > Thanks. The authors have addressed my comments.

---

### Official Review · Reviewer_SXvC · 2023-07-20
**Meaningful benchmark for modeling global fitness landscapes of protein wildtypes in protein engineering**

**Rating:** 5
**Confidence:** 5
**Clarity:** The paper is well-written and accessi…

**Strengths:**

1. This is an interesting work that proposes three new standard datasets and conducts very careful empirical studies.
2. All supervised learning methods use random forest regressors for comparison on downstream tasks, which is a fair way to compare different models.

**Additional Feedback:**

No.

**Correctness:**

The claims about dataset and benchmark seem correct other than the points mentioned above.

**Documentation:**

The authors have provided sufficient detail on data collection and organization, availability and maintenance, and ethical and responsible use, documentation and intended uses. They have also provided URL for access to the dataset, as well as sufficient detail to support reproducibility.

**Ethics:**

No.

**Limitations:**

The authors have adequately addressed the limitations and potential negative societal impact of their work.

**Opportunities For Improvement:**

1. The dataset size is too small, I noticed that the number of sequences in the three datasets are 55, 855 and 615 respectively. I doubt whether these three datasets with such a small number can fully evaluate the ability of pre-trained models and whether these three datasets can be sufficiently representative of the mutation effect prediction tasks. Referring to ProteinGym (https://www.proteinym.org/), which includes nearly a hundred datasets, constructing a representative and sufficient dataset will enable a more comprehensive exploration of the capabilities of the model.
2. Datasets require many partitioning methods. The core problem of mutation effect prediction task is the complex epistasis landscape formed by multi-site mutations. Therefore, in order to assist protein engineering tasks (such as directed evolution), referring to FLIP (https://benchmark.protein.properties/),  *-vs-rest dataset division is necessary.
3. The mutant sequence is generally only a few amino acids different from the wild type sequence, and mean-pooling all residues as sequence representation may not be the most appropriate method. More methods such as extracting sequence representations through convolution or only considering mutant residue information need to be considered. In addition, referring to https://arxiv.org/abs/2011.03443 and https://www.biorxiv.org/content/10.1101/2021.07.09.450648v1.full, the ability of pre-trained models may need to be verified in a zero-shot way, rather than supervised learning methods.
4. In addition, simple sequence representation models such as CNN and structural representation models such as GCN, may be necessary as baselines.

BTW, some minor concerns or typos exist in the paper:
1. In Chapter 3.4, the input of the EVE method is multiple sequence alignments (MSAs), and there will be information leakage. EVE is called a weakly supervised method, which may be more appropriate than zero-shot.
2. In Table 1, GH114 dataset's $N_{tot}$ should be 55.

**Relation To Prior Work:**

Related works are addressed well.

**Summary And Contributions:**

1. This paper studies the generalization ability of three types of representation learning models based on protein sequence, structure and evolution on the task of protein mutation effect prediction.
2. In particular, the authors split datasets into training and testing set based on phylogenetic information, which ensures that no information leakage occurs during the training process.

---

> ### Author Response · Authors · 2023-08-18
> **Rebuttal (part 1)**
>
> We would like to thank the reviewer for their thorough considerations for our paper. We would like to point out that the datasets considered in our benchmark all are composed of highly diverse wildtype proteins which makes direct comparisons to mutation/variant effect prediction dataset and benchmarks difficult due to the inherent differences in size, availability, and diversity.
>
> **Opportunities For Improvement**
> - "*The dataset size is too small, I noticed that the number of sequences in the three datasets are 55, 855 and 615 respectively. I doubt whether these three datasets with such a small number can fully evaluate the ability of pre-trained models and whether these three datasets can be sufficiently representative of the mutation effect prediction tasks. Referring to ProteinGym ([https://www.proteinym.org/](https://www.proteinym.org/)), which includes nearly a hundred datasets, constructing a representative and sufficient dataset will enable a more comprehensive exploration of the capabilities of the model.*"
>
> We agree that the sizes of the datasets for our tasks are small - especially when comparing to mutation effect prediction datasets which often number in the thousands. There is however a distinct lack of published datasets which consider wildtype protein families. While few and small, we believe that our included datasets are representative of what's available and simultaneously reflect the available resources for real-world wildtype discovery campaigns. ProteinGym is a fantastic collection of DMS datasets which makes it possible to evaluate methods for variant effect prediction (which is typically defined as the effect of one or a few mutations). It is unfortunately not possible to curate a collection similar in size and scope for wildtype datasets due the described issues (see below for further elaboration).
>
> - *Datasets require many partitioning methods. The core problem of mutation effect prediction task is the complex epistasis landscape formed by multi-site mutations. Therefore, in order to assist protein engineering tasks (such as directed evolution), referring to FLIP ([https://benchmark.protein.properties/](https://benchmark.protein.properties/)), \*-vs-rest dataset division is necessary.*
>
> We completely agree that \*-vs-rest dataset divisions are very valuable to evaluate mutation effect predictors which can then be used for protein engineering. The FLIP paper did an outstanding job of formalising this procedure for two of their included datasets. Both of these datasets however concern mutation effects.  Since we are working with collections of wildtype proteins, the notion of mutations (either in number or position) does not apply in a similar way since we have no single reference wildtype. To further elaborate on this distinction, we have computed the median pair-wise sequence identity for 48 of the tasks from ProteinGym (akin to the second-to-last column of Table 1 in our paper). Where the overall average for our three datasets is 0.466, the average for the ProteinGym tasks is 0.993. We have added tabulated values for all these datasets in Table A1 and Table A2 (for FLOP and ProteinGym, respectively) in the supplementary materials, which emphasizes the high sequence diversity of the tasks at hand.

---

> ### Author Response · Authors · 2023-08-18
> **Rebuttal (part 2)**
>
> - *The mutant sequence is generally only a few amino acids different from the wild type sequence, and mean-pooling all residues as sequence representation may not be the most appropriate method. More methods such as extracting sequence representations through convolution or only considering mutant residue information need to be considered. In addition, referring to [https://arxiv.org/abs/2011.03443](https://arxiv.org/abs/2011.03443) and [https://www.biorxiv.org/content/10.1101/2021.07.09.450648v1.full](https://www.biorxiv.org/content/10.1101/2021.07.09.450648v1.full), the ability of pre-trained models may need to be verified in a zero-shot way, rather than supervised learning methods.*
>
> We completely agree that mean-pooling residues as sequence representations is not optimal. Given our experience, this is especially the case for mutation effect datasets where the representation differences are small. For wildtype datasets, however, the differences in representations are more pronounced leading to more distinct signals. We have now added a reference to a recent paper in which alternatives to mean-pooling are proposed, e.g. by training a bottleneck model or using concatenation approaches (see https://www.nature.com/articles/s41467-022-29443-w). These alternatives are however demonstrated using mutational effect dataets (i.e. of lower diversity). Using such approaches for the diverse datasets we consider is non-trivial and considered out of the scope for the benchmark at hand, in which we present results from common methods.
>
> We also agree that zero-shot approaches are necessary to consider. Our original paper contained comparisons to three zero-shot methods. Additionally, we have now included results ProteinMPNN, which outperforms the initial three models. The supervised approaches given embeddings from pre-trained models however still consistently outperforms the zero-shot estimators.
>
> - *In addition, simple sequence representation models such as CNN and structural representation models such as GCN, may be necessary as baselines.*
>
> While we agree that more simple baselines would be interesting and relevant to consider, manually pre-training representation models is considered out of the scope of this paper. This would require constructing self-supervised learning strategies, curating sufficiently sized datasets, as well as significant computational resources. For a representation using a CNN approach, we can refer to MIF-ST, where the ST (sequence transfer) component makes use of a CNN-based pLM. For GCN, we refer to both MIF-ST and ESM-IF1 which use GNNs as a core way of handling structure inputs.
>
> An alternative would be to train an end-to-end predictor, which takes sequences/structures as inputs and predicts the regression targets directly. While more feasible than training representation models, this clashes with our representation-centric approach, where we wish to evaluate the efficacy of representations in a transfer-learning setting. Such an approach would thus have an unfair advantage where the representations are directly shaped using the target values. Additionally, given the limited sizes of our datasets, it is by no means guaranteed that these models which due to the complexity of the input data would have a large number of parameters would converge in any meaningful way.
>
> - "*In Chapter 3.4, the input of the EVE method is multiple sequence alignments (MSAs), and there will be information leakage. EVE is called a weakly supervised method, which may be more appropriate than zero-shot.*"
>
> We agree that using MSAs that have been generated using the sequences of interest constitutes a type of information leakage. This is similarly the case for using pre-trained models like the ESM-suite, AlphaFold2, MIF-ST, and so on - as a number of the wildtype sequences we are actively considering have possibly been included in the training sets for these models. While this is the case, we do not consider it to be a fatal source of data-leakage. However, we agree that this is necessary to discuss and now include a discussion section accordingly in section 3 in the paper. We thank the reviewer for highlightning this point.
> We would not consider EVE to be a weakly-supervised model, however. Our understanding is that weak-supervision requires some notion of utilising or synthesizing the data *labels* during training as a way of circumventing the general labelling-scarcity issue (https://arxiv.org/abs/2202.05433). We train EVE in a completely unsupervised way as a VAE, where no labels are involved.
>
> - "*In Table 1, GH114 dataset's $N_\text{tot}$ should be 55.*"
>
> Good catch - thank you for noticing and making us aware!

---

> > ### Comment · Reviewer_SXvC · 2023-08-30
> > **Response to rebuttal**
> >
> > Thanks to the authors for addressing my concerns in their rebuttal. However, there is still one concern related to the dataset and evaluation tasks. As a benchmark work, the three datasets which have 55, 855 and 615 samples respectively are insufficient. The authors say the datasets are representative, although they are small. Nevertheless, there are no in-depth explanations about why the small datasets are enough and representative, and what are the real challenges in the tasks of fitness landscapes of protein wildtypes. Apart from this, the responses have addressed other concerns. As a result, I update my score from 4 to 5.

---

> > > ### Author Response · Authors · 2023-08-30
> > > **Reponse to reviewer**
> > >
> > > We are grateful for your response and the score increase!
> > >
> > > We agree that more and larger datasets would be welcome additions to the benchmark. The current literature does however not offer many (nor large) wildtype fitness landscapes for single protein families. We are aware that these requirements limit the number of applicable datasets, yet these constraints are what characterize the challenge of wildtype discovery we're seeking to investigate and are thereby intrinsic to the domain.
> > >
> > > While we mentioned data-scarcity issues (L20-30) and current benchmark limitations for wildtype discovery (L85-90) and reflected on these issues in the conclusion (L345-353), we have now (in the most recent revision) added further elaborations in the Related work section (L91-98) and the Datasets section (L192-195), where we now acknowledge the issue further and emphasize why the curated datasets are representative and relevant.
> > >
> > > We agree that this could have been stated more clearly and we appreciate your feedback.

---

### Official Review · Reviewer_aTwL · 2023-07-20
**A Timely Wildtype Discovery Benchmark for Protein Engineering**

**Rating:** 7
**Confidence:** 4

**Strengths:**

* The authors' proposed dataset splits for the `GH114`, `CM`, and `PPAT` tasks are reasonable and soundly constructed.
* The authors' initial selection of baseline methods for comparison in the proposed benchmarks represents several well-known and recent sequence and structure-based advances in protein representation learning.
* The authors' analysis of their curated datasets and splits reveals the importance (for protein engineering data) of ensuring that such splits are not random yet do not prevent methods from leveraging subfamily information to regress a target value.
* The ablation studies the authors include are insightful in that they clearly explain the authors' line of reasoning in reporting the results that they do in the main text.
* The datasets, model source code, and tasks are all well documented on GitHub.

**Additional Feedback:**

This reviewer would like to thank the authors for highlighting the importance of the initial wildtype discovery phase in ML-based protein engineering. Such work will likely have a measurable impact on the field going forward.

**Clarity:**

* The manuscript is written in a way that makes it relatively easy for readers to closely follow the authors' narrative and chains of reasoning.

**Correctness:**

* The claims made throughout the manuscript are correct. Similarly, the authors' proposed dataset splits, benchmark tasks, and evaluation methods are all reasonable.

**Documentation:**

* The authors' curated datasets, benchmark tasks, and methods are all well-documented. The work appears to be reproducible based on their documentation in the attached GitHub repository.

**Ethics:**

* No ethical concerns are raised by this work.

**Limitations:**

* It is clear that the authors have carefully noted the limited size of their curated datasets, and such discussions are well grounded in the real-world challenges of collecting high-quality (i.e., low-noise) catalytic or functional annotations of proteins using modern assays. Nonetheless, advances in protein representation learning methods may only marginally increase performance for the authors' proposed benchmark without introducing additional fine-tuning data related to the tasks at hand (or without adding more high-quality data to each method's training datasets). As such, without also incorporating a discussion on the real-world utility of transfer learning and active learning within the context of the authors' proposed benchmark (or perhaps outlining for readers some next steps in these directions), the impact of this new benchmark for machine learning practitioners may be unnecessarily limited.

**Opportunities For Improvement:**

* The authors' baseline methods are void of many recent advances in protein graph representation learning, including new equivariant neural network architectures for learning from protein point clouds (e.g., ProteinMPNN, GVP-GNNs, ProNet, CDConv, etc). Given that the authors have already predicted structures for each of the datasets' sequences, it should be relatively simple for the authors to include such new methods in their existing benchmark.
* Regarding the placement of Figure 2, it may make more sense to place it on Page 4 instead of 3 so that (by then) readers will have already been introduced to the topic (and importance) of data splitting.

**Relation To Prior Work:**

* The authors clearly discuss their work's relation to prior works. This new work fits nicely within the context of the existing ML-based protein engineering literature.

**Summary And Contributions:**

The authors propose a rigorously-constructed and timely computational benchmark for wildtype protein sequence discovery. They include several well-known protein representation methods in the proposed benchmark, highlighting that there is (currently) no best-performing representation modality overall for the proposed regression tasks (which is useful knowledge in and of itself). Especially encouraging to see is the authors' inclusion of well-motivated ablation studies demonstrating the importance of carefully constructing one's protein engineering dataset splits. Overall, this work seems to be a good fit for this year's proceedings, given the soundness of this work in addition to recent advances and interest in ML-driven protein engineering efforts.

---

> ### Author Response · Authors · 2023-08-18
> **Rebuttal**
>
> We like to thank the reviewer for their thorough review as well as their interest in - and appreciation of - our work on wildtype discovery. We will touch upon the raised points in turn.
>
> **Improvements**
>
> - *The authors' baseline methods are void of many recent advances in protein graph representation learning, including new equivariant neural network architectures for learning from protein point clouds (e.g., ProteinMPNN, GVP-GNNs, ProNet, CDConv, etc). Given that the authors have already predicted structures for each of the datasets' sequences, it should be relatively simple for the authors to include such new methods in their existing benchmark."*
>
> It is certainly fascinating to follow the rapid advances in protein-focused machine learning. Actually, our original study did include one of these methods, the GVP-GNN - although under the name ESM-IF1. We now provide both names in the paper to avoid confusion. Furthermore, we now also include zero-shot results for ProteinMPNN and can conclude that it outperforms the other zero-shot predictors present in our benchmark. We thank you for the suggestion.
>
> - *Regarding the placement of Figure 2, it may make more sense to place it on Page 4 instead of 3 so that (by then) readers will have already been introduced to the topic (and importance) of data splitting.*
>
> We agree and have moved the figure accordingly - thank you for the suggestion.
>
> **Limitations**
>
> - *It is clear that the authors have carefully noted the limited size of their curated datasets, and such discussions are well grounded in the real-world challenges of collecting high-quality (i.e., low-noise) catalytic or functional annotations of proteins using modern assays. Nonetheless, advances in protein representation learning methods may only marginally increase performance for the authors' proposed benchmark without introducing additional fine-tuning data related to the tasks at hand (or without adding more high-quality data to each method's training datasets). As such, without also incorporating a discussion on the real-world utility of transfer learning and active learning within the context of the authors' proposed benchmark (or perhaps outlining for readers some next steps in these directions), the impact of this new benchmark for machine learning practitioners may be unnecessarily limited.*
>
> The reason that the benchmark is relevant, despite these concerns, is that there is a trade-of between the experimental effort spent in wildtype exploration vs. experimental effort spent in subsequent protein engineering, in essence an exploration/exploitation tradeoff. Often, only limited resources are allocated to wildtype exploration, since it is inherently wasteful. This is unlikely to change, unless we find ways to improve our wildtype search strategy, which will require better predictions - either based on small datasets or using zero-shot prediction. We therefore consider the limited dataset sizes as an inherent condition and limitation in this domain. However, as the reviewer suggests, one can imagine fine-tuning representations on the protein family of interest - or developing weakly supervised general-purpose models that incorporate relevant properties that correlate with the function of interest (e.g. predictions of thermal stability) - either boosting small-dataset regression or zero-shot prediction on this problem. It is also important to remember that these tools are actually used on a daily basis, and any (even small) improvement will have have real-world impact. We have now included the above reflections in the conclusion (which has been rewritten) as it serves to highlight an important potential limitation in the expected improvements to the proposed benchmark tasks. We thank the reviewer for their insights and suggestions and believe that the validity of benchmark is improved as a result thereof.

---

> > ### Comment · Reviewer_aTwL · 2023-08-21
> > **Response to Rebuttal**
> >
> > I would like to thank the authors for addressing my remaining concerns through their rebuttal. It is interesting to see how graph-based methods such as ProteinMPNN compare to the original baselines included in this work. In the future, I would encourage the authors to consider adding even more graph-based methods to their benchmark results (e.g., CDConv, ProNet, etc.) to see if these strong trends (in protein graph representation learning) hold.

---

### Official Review · Reviewer_XdTc · 2023-07-21
**FLOP: Tasks for Fitness Landscapes Of Protein wildtypes**

**Rating:** 4
**Confidence:** 3

**Strengths:**

The authors have designed an Python API in away that abstracts out most of the backend necessary for obtaining and processing the dataset. The authors have shared the scripts they used for generating benchmark dataset, evaluations and key statistical analyses. They also provide guidance on how readers may apply and extend their python API by sharing tutorial on how to add new protein representations.

**Additional Feedback:**

N/A

**Clarity:**

In section through 4 through 8, I found some sentences appeared to be incomplete or lack in necessary detail or supporting evidence. In certain cases, the selection of specific words in the sentences seems inappropriate or the sentences are composed of multiple statements that may better separate into stand-alone for the sake of clarity. Here are some of the examples:
- " We concretely use “GraphPart” [21]." what do authors mean by "concretly"?
- " The three curated datasets and the corresponding fitness landscapes are here motivated and described. " - What do the authors mean by "motivated" here?
- "Protein language models (pLMs) that are trained on hundreds of millions of protein sequences in an unsupervised fashion have been proven to be competitive representations for a multitude of tasks [23–25]." : What are these tasks?
- "However, purifying enzymes requires significant work, resulting in a limited number of tested sequences, but of higher experimental quality"  -  What do the authors intend to convey in this sentence?
- "The stratification is achieved by creating a binary label which indicates whether a protein has low or high target value,    e.g., by fitting a two-component Gaussian mixture model" - Did the authors trained two-component Gaussian mixture model on the top of the raw target values? Detail on this data preparation is not provided in the main body (or link to appropriate supplementary information)
- "Accurately identifying enzymes with the highest activities towards a specific substrate is of central importance during enzyme engineering. This requires that assay observations are directly comparable , which includes ensuring identical experimental assay conditions, such that enzymes  are evaluated at identical concentrations and purity levels“: Each of the statement includes multiple components in one sentence.  For example, I'd re-write the latter sentence to "To achieve this, it is essential to ensure that assay observations are directly comparable. This involves maintaining identical experimental assay conditions, including evaluating enzymes at the same concentrations and purity levels."
 - “Often, a family-wide, one-hot encoded MSA proved competitive, highlighting the difficulty of creating efficient representations. We encourage the design of new representations which can avoid such pooling operations as were applied  to the pretrained model embeddings, which are bound to filter out potentially important information": What do the authors mean by "highlighting the difficulty of creating efficient representations" here?  Also, for the second sentence do the authors intend to communicate " We encourage the development of novel representations that can bypass the pooling operations used in the pretrained model embeddings, as these operations may filter out valuable information" ?
- "For the PPAT dataset, we see the predictive performance
 when using repeated random splitting instead of stratified, homology-based splitting. " - The sentence need to be re-written for clarity



**Correctness:**

The paper stands on the assumption that 1) existing benchmarks are either insufficient or inappropriate to evaluate the models predicting properties of wildtype sequences and 2) WT sequences varies substantially more in sequence landscape that what's captured in current benchmarks (e.g. mutational scanning experiments)
- The important details regarding the dataset is missing in a way that provides little support for what the authors aims to investigate. Specifically, in Section 4 where the authors discuss benchmark dataset, it's unclear whether the dataset - GH114, CM, PPAT - is consisted of wildtype sequences. For example, it's mentioned that CM dataset includes artificial sequence generated from Monte Carlo simulation. However, the logical link of why including the aritificial sequences is still valid and necessary to test the authors' hypotheses is not provided. Additionally, for each of the dataset, it's unclear whether the observed sequence identity for enzymes are lower than existing mutational fitness landscape dataset without any baseline measurements/references provided.
- I did not quite follow the authors' logic on ablation study for regression on both activate and inactive sequences. The authors should follow-on what they mean by target modality and whether including target modality is introducing spurious correlation (and therefore need to be removed)


**Documentation:**

Overall, the paper can benefit from additional information regarding the provenance of dataset. While authors have provided simple summary statistics for the benchmarks, other crucial information necessary for utilizing and applying the data is missing: namely, where the data has come from: data collection, organization; where the data may be accessed: availability, hosting, licensing and plan for the maintenance.

**Limitations:**

Authors have not addressed the limitations/potential negative societal impact of their work. It appears that one of the benchmark dataset contains the model -generated sequences.  Due to the lack of details on these sequences are obtained, it's unclear to me how the authors deal with common challenges involved with using model-generated data for training.  I hope that follow-up writing on this section clarifies some of the questions I have.

**Opportunities For Improvement:**

Here are some of the suggestions that I have:

- In section 4.1 through 4.3, clearly state whether the dataset is consisted only of Wild type sequences. In addition, the authors should provide further logical connection as to why they include artificial sequences generated from Monte Carlo and demonstrate that generated sequences closely match that of the real sequence or how they filter out unrealistic sequences.

- In Table 1 and Section 4.4 ,  it's unclear whether the observed sequence identity for benchmarks are meaningfully lower than existing mutational fitness landscape dataset without any baseline measurements/references provided. The authors should provide further analyses to show that wildtype sequence landscape is substantially varied than what's currently captured in existing mutational scanning experiments.

- I had difficulty understanding section 4 and 5. For example: “We encourage the design of new representations which can avoid such pooling operations as were applied  to the pretrained model embeddings, which are bound to filter out potentially important information “: The authors should provide further analyses demonstrating that this is indeed the case or references supporting the idea.

- For Section 6, I didn't understand the difference between hold-out validation and Random partitioning. Hope the authors make it clear how two are different from each other and which of them related to the point the authors have made in Figure 2.

- Improve the quality of writing[elaborated in Clarity section]: The paper would benefit from follow-up editorial/proof-reading efforts focusing on correcting grammatical errors and improving the overall clarity and coherence. Some sentences read as incomplete or left to be further constructed with more details, supporting paragraph or reference.



**Relation To Prior Work:**

The authors provide sufficient information regarding the related works, such as FLIP benchmark, CASP benchmark, TAPE benchmark and so on.

**Summary And Contributions:**

The paper presents a curated benchmark dataset for initial wildtype protein discovery. It investigates the predictive power of different protein representations, such as language model-based, structure-based, and evolution-based representations. The authors share the results of a few analyses that underscores the importance of coherent split strategies to avoid overly optimistic estimates. They share their results and scripts in public repository.

---

> ### Author Response · Authors · 2023-08-18
> **Rebuttal (part 1)**
>
> We would like to thank the reviewer for their constructive and thorough feedback and considerations. We will address the raised points in turn several blocks due to character limits.
>
> **Opportunities For Improvement**
>
> - *In section 4.1 through 4.3, clearly state whether the dataset is consisted only of Wild type sequences. In addition, the authors should provide further logical connection as to why they include artificial sequences generated from Monte Carlo and demonstrate that generated sequences closely match that of the real sequence or how they filter out unrealistic sequences.*
>
> We acknowledge that our communication regarding the preprocessing of the CM dataset could be considered lacking. We have now addressed the motivation to include/exclude parts of the CM dataset in detail in the supplementary materials (see section A.3.1 specifically).
>
> The designed sequences from the CM dataset are obtained by Monte Carlo sampling via Boltzmann-machine learning direct coupling analysis (bmDCA). The sequences sampled at temperatures $T=0.33$ and $T=0.66$ match the empirical first-, second-, and higher-order statistics of the natural homologs, thereby highlightning their similarity in sequence space.
> These sequences also exhibit comparable catalytic levels compared to the natural homologs when experimentally synthesized.
> Given the similarity to the natural homologs in both sequence and expression, the sequences have been included.
>
> The sequences sampled at temperature $T=1$ and the sequences designed using a simple profile model (where amino acids were only sampled according to position-specific conservation, i.e., first-order statistics only) were discarded. The high-temperature sequences were almost exclusively non-functional while also being too distant from the wildtype homologs: the mean sequence identity to each sequence's nearest natural homolog was 0.55. For comparison, the mean sequence identity to nearest natural homologs for the sampled sequences at temperatures 0.33 and 0.66, is 0.81 and 0.76, respectively. While the sequences sampled using the profile model were similar in first-order statistics by design (mean sequence identity of 0.76 to nearest homologs), the sequences were exclusively non-functional. These arguments and statistics are now provided in section A.3.1 in the supplementary materials.
>
> For completeness, we now include the benchmark results when only the natural homologs were used for the CM dataset (thus computing new stratification boundaries and slitting the dataset anew). These results can be seen in Figure A14 in the supplementary materials. We note that the pattern is similar to the main benchmark results, while the scores however generally are lower and more variable (which is unsurprising given the reduced dataset size).
>
> - *In Table 1 and Section 4.4 , it's unclear whether the observed sequence identity for benchmarks are meaningfully lower than existing mutational fitness landscape dataset without any baseline measurements/references provided. The authors should provide further analyses to show that wildtype sequence landscape is substantially varied than what's currently captured in existing mutational scanning experiments.*
>
> We have now computed similar summary statistics for 48 datasets from the ProteinGym benchmark. The average median sequence identity for these datasets is $> 0.99$, with a standard deviation of $0.001$. For comparison, the average median sequence identity for the FLOP datasets is $0.466$ with a standard deviation of $0.067$, implying that the DMS data from ProteinGym is nearly identical in sequence space (since it is often separated by only single) while the sequences in FLOP are less than 50 % similar. We have included two additional tables in the supplementary materials (see Table A1 and Table A2), which show these quantities for the datasets from FLOP and ProteinGym, respectively.
> We have also incorporated this information in the article as it highlights the stark difference in diversity between DMS data and the datasets from the FLOP benchmark. We thank you for the suggestion.

---

> ### Author Response · Authors · 2023-08-18
> **Rebuttal (part 2)**
>
> - *I had difficulty understanding section 4 and 5. For example: “We encourage the design of new representations which can avoid such pooling operations as were applied  to the pretrained model embeddings, which are bound to filter out potentially important information “: The authors should provide further analyses demonstrating that this is indeed the case or references supporting the idea.*
>
> We have added a reference to a recent article in which the authors demonstrate alternative approaches by, e.g., training a bottleneck model to achieve fixed-sized embeddings and thereby reach better performance (see https://www.nature.com/articles/s41467-022-29443-w). The example considered in the aforementioned article however uses aligned protein sequences (i.e. of equal length), which adds additional overhead to method at hand. For benchmarking purposes, we chose to keep aligned sequences and protein language model embeddings separate.
>
> - *For Section 6, I didn't understand the difference between hold-out validation and Random partitioning. Hope the authors make it clear how two are different from each other and which of them related to the point the authors have made in Figure 2.*
>
> We agree that our revision process prior to submission had resulted in too compact descriptions of our ablation studies. We have now expanded/altered each of the three subsections to more clearly describe the ablations and their implications.
>
> "Random partitioning" refers to creating three partitions by randomly sampling sequences for each, and then evaluating the representations using cross-validation. We ran this experiment a total of three times using three different seeds. This experiment thereby has no consideration of sequence identity or potential leakage from similar sequences.
>
> "Hold-out validation" refers to training the random forest regressor for each representation only once using one dataset partition for training, one for validation, and the last for testing. The distinction to random partitioning (apart from training on separate partitions vs. training only once) lies in how the partitions are generated. For the hold-out experiment, we used the benchmark partitions, i.e. ensuring that sequences in different partitions can only be equal up to a certain threshold and that the target values are stratified.  Due to differences in the partitions (by design), the accuracy of the predictive model might be different when trained and evaluated on different partitions. Hold-out validation can therefore lead to biased results where we get an inaccurate impression of how well a representation performs. By running the CV-loop, we get a nuanced and more trustworthy impression of the results - especially via the errorbars which show that the performance can vary between splits.
>
> We refer to the respective sections in the updated article.
>
>
>
> - *Improve the quality of writing\[elaborated in Clarity section\]: The paper would benefit from follow-up editorial/proof-reading efforts focusing on correcting grammatical errors and improving the overall clarity and coherence. Some sentences read as incomplete or left to be further constructed with more details, supporting paragraph or reference.*
>
> We will touch upon this in the clarity section below.
>
> **Limitations**
>
> - *Authors have not addressed the limitations/potential negative societal impact of their work. It appears that one of the benchmark dataset contains the model -generated sequences. Due to the lack of details on these sequences are obtained, it's unclear to me how the authors deal with common challenges involved with using model-generated data for training. I hope that follow-up writing on this section clarifies some of the questions I have.*
>
> We agree that using model-generated data for training is traditionally ill-advised if the data is inherently synthetic. We would however like to emphasize that while a subset of the CM sequences are indeed model-generated, the sequences were subsequently synthesized and measured leading to the activity assay values. The dataset is thereby entirely composed of sequence/target pairs which have been observed in a wet-lab and thus includes no synthetic data. For additional details on our choice of including a subset of these sequences, we refer to our response in the improvements section. We agree with the reviewer that this is a crucial point to discuss and we hope that our revised article more explicitly reflects our considerations and choices.

---

> ### Author Response · Authors · 2023-08-18
> **Rebuttal (part 3)**
>
> **Correctness**
> - *The important details regarding the dataset is missing in a way that provides little support for what the authors aims to investigate. Specifically, in Section 4 where the authors discuss benchmark dataset, it's unclear whether the dataset - GH114, CM, PPAT - is consisted of wildtype sequences. For example, it's mentioned that CM dataset includes artificial sequence generated from Monte Carlo simulation. However, the logical link of why including the aritificial sequences is still valid and necessary to test the authors' hypotheses is not provided. Additionally, for each of the dataset, it's unclear whether the observed sequence identity for enzymes are lower than existing mutational fitness landscape dataset without any baseline measurements/references provided.*
>
> We hope that our above discussions of the inclusion of model-generated sequences as well as our inclusion of summary statistics for DMS datasets to highlight the diversity of the wildtype datasets in the Improvements and Limitations sections are satisfactory.
>
> - *I did not quite follow the authors' logic on ablation study for regression on both activate and inactive sequences. The authors should follow-on what they mean by target modality and whether including target modality is introducing spurious correlation (and therefore need to be removed)*
>
> We have now elaborated on and clarified the motivation behind the ablation study. By target modalities we simply mean a target value for regression where the values are likely to belong to one of several major clusters, e.g. belonging to a bimodal distribution. For the full CM dataset, approximately half of the proteins have activity values centred on 0, while the the other half has values centred on 1 (this can be seen in Figure A1 in the supplementary materials). While the regression model is able to capture a distinction between the two modes, the within-mode accuracy is significantly lower (as reflected by our benchmark results). We have made clarifications in the paper and we thank the reviewer for highlighting the potential issue.
>
> **Clarity**
>
> We would like to thank the reviewer not just for their thorough suggestions for where to rewrite but also for coming with valuable suggestions for how to rephrase the unclear sentences.
>
> - *" We concretely use “GraphPart” \[21\]." what do authors mean by "concretly"?*
>
> We have removed the word concretely for clarity.
>
> - *" The three curated datasets and the corresponding fitness landscapes are here motivated and described. " - What do the authors mean by "motivated" here?*
>
> The motivation sections for each dataset seek to describe why the task and dataset at hand is interesting and thereby relevant for benchmarking. We have altered the motivation sections to more clearly describe and motivate the tasks at hand.
>
> - *"Protein language models (pLMs) that are trained on hundreds of millions of protein sequences in an unsupervised fashion have been proven to be competitive representations for a multitude of tasks \[23–25\]." : What are these tasks?*
>
> We have added examples of these tasks to the article: "Protein language models (pLMs) that are trained on hundreds of millions of protein sequences in an unsupervised fashion have been proven to be competitive for a multitude of tasks including supervised prediction of protein properties, residue contact prediction, variant effect prediction, etc."
>
> - *"However, purifying enzymes requires significant work, resulting in a limited number of tested sequences, but of higher experimental quality" - What do the authors intend to convey in this sentence?*
>
> We agree that our initial phrasing was unclear and have changed it accordingly. The section now reads:
>
> "Accurately identifying enzymes with the highest activities towards a specific substrate is of central importance during enzyme engineering. To achieve this, it is essential to ensure that assay observations are directly comparable. This includes maintaining identical experimental assay conditions, including evaluating enzymes at the same concentrations and purity levels. However, purifying enzymes requires significant work and resources, often resulting in assays composed of fewer sequences, which are in turn of higher experimental quality."
>
>
> - *"The stratification is achieved by creating a binary label which indicates whether a protein has low or high target value,    e.g., by fitting a two-component Gaussian mixture model" - Did the authors trained two-component Gaussian mixture model on the top of the raw target values? Detail on this data preparation is not provided in the main body (or link to appropriate supplementary information)*
>
> We now describe the stratification procedure for each dataset in the supplementary materials (including the dataset-specific thresholds), where each dataset now has a "Stratification threshold" paragraph. We thank the reviewer for noticing the lack of these values in the initial submission.

---

> ### Author Response · Authors · 2023-08-18
> **Rebuttal (part 4)**
>
> - *"Accurately identifying enzymes with the highest activities towards a specific substrate is of central importance during enzyme engineering. This requires that assay observations are directly comparable , which includes ensuring identical experimental assay conditions, such that enzymes  are evaluated at identical concentrations and purity levels“: Each of the statement includes multiple components in one sentence. For example, I'd re-write the latter sentence to "To achieve this, it is essential to ensure that assay observations are directly comparable. This involves maintaining identical experimental assay conditions, including evaluating enzymes at the same concentrations and purity levels."*
>
> We prefer your phrasing and have included it accordingly - see our response above for the full section.
>
> - *“Often, a family-wide, one-hot encoded MSA proved competitive, highlighting the difficulty of creating efficient representations. We encourage the design of new representations which can avoid such pooling operations as were applied  to the pretrained model embeddings, which are bound to filter out potentially important information": What do the authors mean by "highlighting the difficulty of creating efficient representations" here? Also, for the second sentence do the authors intend to communicate " We encourage the development of novel representations that can bypass the pooling operations used in the pretrained model embeddings, as these operations may filter out valuable information" ?*
>
> We have changed the first sentence to the following for clarity:
>
> "Often, a family-wide, one-hot encoded MSA proved competitive, highlighting the difficulty of creating informative, machine learning-based representations."
>
> For the second sentence, we prefer yours and have changed ours accordingly.
>
> - *"For the PPAT dataset, we see the predictive performance when using repeated random splitting instead of stratified, homology-based splitting. " - The sentence need to be re-written for clarity*
>
> During our clarification of the ablation study, the referenced sentence has been rewritten for clarity.
>
> **Documentation**
> - *Overall, the paper can benefit from additional information regarding the provenance of dataset. While authors have provided simple summary statistics for the benchmarks, other crucial information necessary for utilizing and applying the data is missing: namely, where the data has come from: data collection, organization; where the data may be accessed: availability, hosting, licensing and plan for the maintenance.*
>
> We believe that the supplementary materials and the GitHub repository provide detailed information on how to access and use the benchmark as well as details on hosting and licensing. For details, see the section "Dataset details" in the supplementary materials, where we describe the overall process for how to access and use the data, and where we provide a subsection for each dataset in which we carefully describe exactly where the data comes from, and how we processed it. For additional details on access, licensing, and maintenance, we refer to the section titled "Mandatory dataset information details" in the supplementary materials. If the reviewer has specific suggestions for improvements, we would be happy to make changes accordingly.

---

### Author Response · Authors · 2023-08-30
**Revision summary**

Dear Reviewers,

We would again like to thank you all for taking time to review our work. We have received valuable feedback and have revised our manuscript (and supplementary materials) accordingly. We are happy to engage in further discussions if any new questions have arisen. We here provide a non-comprehensive summary of the changes we have made. Each point is described in detail in the individual rebuttals.

- Updated conclusion which reflects current limitations and possible future steps of ML-driven wildtype discovery (aTwL, YyxM)
- Inclusion of ProteinMPNN as zero-shot predictor (aTwL)
- Further details on the curation of the CM dataset and reasoning behind including model-generated sequences (XdTc)
- Table and discussion of observed mean sequence identities in ProteinGym benchmark for comparison (XdTc, SXvC)
- Clearer descriptions and discussions of ablation studies (XdTc)
- Various changes to phrasing/typo fixes (XdTc)
- Discussion of mean-pooling for sequence representations with appropriate citation (SXvC)
- Restructuring/expansion of dataset sections in supplementary materials (XdTc)

As the discussion deadline is fast approaching, we would be grateful if the reviewers would take our revisions into consideration and reach out in case of follow-up questions and clarifications.

Kind regards,

the Authors.

---

### Decision · Program_Chairs · 2023-09-22

**Decision:**

Reject

**Comment:**

This paper presents a benchmark dataset for evaluating if wildtype (naturally occuring) proteins are suitable for a task.

The authors study how the choice of the representation affects the downstream predictive performance, which is very reasonable.

It is surprising that a random forest regression is the best model they find, suggesting that additional side-information for representation learning was not successfully used.

Overall it seems that the introduced datasets are too small to get raise broad interest in the ML community. This combined with the poor presentation make this paper not ready for publication.